# A Reparameterized Discrete Diffusion Model for Text Generation

## Abstract

This work studies discrete diffusion probabilistic models with applications to natural language generation. We derive an alternative yet equivalent formulation of the sampling from discrete diffusion processes and leverage this insight to develop a family of *reparameterized discrete diffusion models*. The derived generic framework is highly flexible, offers a fresh perspective of the generation process in discrete diffusion models, and features more effective training and decoding techniques. We conduct extensive experiments to evaluate the text generation capability of our model, demonstrating significant improvements over existing diffusion models.

## 1 Introduction

Diffusion-based generative models (Sohl-Dickstein et al., 2015; Ho et al., 2020; Song et al., 2021b), or diffusion models for short, have achieved remarkable progress and shown great success in generating high-quality and photo-realistic images (Ramesh et al., 2022; Saharia et al., 2022; Rombach et al., 2022; Balaji et al., 2022; Peebles & Xie, 2022). Researchers have successfully extended diffusion models to various data modalities beyond 2D images, including audio (Kong et al., 2021), video (Ho et al., 2022), as well as molecule generation (Hoogeboom et al., 2022b; Jo et al., 2022). There has also been a surge of interest in extending diffusion models to natural languages (Hoogeboom et al., 2021; Austin et al., 2021; Li et al., 2022b; Dieleman et al., 2022). Diffusion-based language models are appealing in that the generation process is typically conducted in a non-autoregressive manner, which features in-parallel decoding by design and potentially a faster runtime (Hoogeboom et al., 2021; Austin et al., 2021). In addition, due to the iterative reconstruction process in diffusion-based models, it is often possible to refine the previously generated texts (Savinov et al., 2022). As a result, compared with the conventional auto-regressive models, diffusion-based language models are more flexible and attain better trade-offs between generation quality and efficiency.

However, there are noticeably fewer success cases in employing diffusion models for large-scale text generation tasks. This is possibly due to the discrete nature of natural languages, while most conventional diffusion models focus on continuous-valued contents. To bridge the discrepancy, a recent line of work suggests conducting continuous diffusion processes over token embeddings (Li et al., 2022b; Gong et al., 2022; Strudel et al., 2022; Dieleman et al., 2022) or logits (Han et al., 2022; Richemond et al., 2022). Nevertheless, these approaches often require designing a well-crafted rounding scheme to convert the diffused continuous vectors to the actual discrete tokens. In addition, existing continuous diffusion models often require a large number of sampling iterations to achieve the desired performance. This issue is exacerbated in the case of modeling texts, as the diffusing steps over text embeddings are hard to be translated to significant movements of token states due to the rounding quantization. This results in a considerably slower runtime than the auto-regressive language models. For example, a recent continuous text diffusion model (DiffuSeq; Gong et al., 2022) runs several orders of magnitude slower than the auto-regressive baseline of a similar scale, as shown in Figure 2b. Different from the above, another research direction focuses on diffusion processes that directly operate on discrete state spaces (Sohl-Dickstein et al., 2015; Hoogeboom et al., 2021; Austin et al., 2021, see §2). However, they are relatively under-explored and often achieve inferior results in text generation.

In this work, we demonstrate that discrete diffusion models can serve as strong baselines for text generation. By re-examining the formulation, we observe that sampling from discrete diffusion

models admits a novel yet equivalent *reparameterization* (§3). Specifically, starting with a completely noisy sequence, the sampling procedure in a discrete diffusion model is equivalent to the following *route-and-denoise* process where at each iteration, each token within the sequence is either denoised or reset to noisy states according to an underlying stochastic *routing* mechanism (§3.2). The router assigns the same probability to the decision for each token, processing the sequence in a uniform manner. Based on this insight, we propose **R**eparameterized **D**iscrete diffusion **M**odels (RDMs; Figure 1), a new family of models that respects the reparameterization and formulates the routing process explicitly. RDMs enjoy appealing properties for both training (§4.2) and sampling (§4.3): (1) *Simplified training*. We demonstrate that the training objective for RDMs can be reduced to a re-weighted standard cross-entropy loss. Furthermore, the loss objective is invariant to different routing probabilities up to reweighting, indicating that the large family of RDMs with distinct routing processes can be trained with the same surrogate objective; (2) *Flexible sampling*. The shared training objective makes sampling highly flexible and allows more expressive routing processes. In particular, we develop an adaptive routing strategy that routes tokens to the denoised state only if the router outputs high scores instead of uniformly processing all the tokens. Equipped with such training and decoding schemes, we demonstrate that RDMs significantly improve vanilla discrete diffusion models across several standard text generation benchmarks (§5). They also achieve much better performance than the continuous diffusion models while running several orders faster.

## 2 BACKGROUND

Let $\mathbf{x}_0 \sim p_{\text{data}}(\mathbf{x}_0)$ denote a discrete random variable with $K$ possible outcomes. To ease notation, we represent discrete variables as one-hot vectors in $\{0, 1\}^K$, which is 0 everywhere except that the entry corresponding to the current state is 1. Discrete diffusion probabilistic models (Sohl-Dickstein et al., 2015; Hoogeboom et al., 2021; Austin et al., 2021) are usually defined as a class of latent variable models characterized by a forward and backward process. The *forward* process aims to gradually transform input data to some noise distribution $q_{\text{noise}}$ through $T$ intermediate latent variables $\mathbf{x}_1, \ldots, \mathbf{x}_T \in \{0, 1\}^K$, with the forward transition $q(\mathbf{x}_t|\mathbf{x}_{t-1}) = \beta_t\mathbf{x}_{t-1} + (1 - \beta_t)q_{\text{noise}}$. In this case, the distribution $q(\mathbf{x}_t|\mathbf{x}_0)$ is available in closed form,

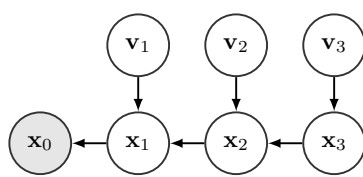

(a) *Conventional* discrete diffusion.

(b) *Reparameterized* discrete diffusion.

$$q(\mathbf{x}_t|\mathbf{x}_0) = \alpha_t\mathbf{x}_{t-1} + (1 - \alpha_t)q_{\text{noise}}, \qquad (1)$$

Figure 1: Graphical models of backward diffusion process.

where $\alpha_t := \prod_{i=1}^{t}\beta_i$ is specified to decrease from 1 to 0 w.r.t. $t$. $q_{\text{noise}}$ characterizes different diffusion processes; for example, **multinomial diffusion** (Hoogeboom et al., 2021) adopts a uniform noise distribution over the vocabulary $\{1, 2, \ldots, K\}$; alternatively, **absorbing diffusion** specifies $q_{\text{noise}}$ to be a point mass with all of the probability on an absorbing state (Austin et al., 2021).

The *backward* process $q(\mathbf{x}_{t-1}|\mathbf{x}_t)$ is the key ingredient for diffusion-based generative modeling, based on which we can start with unstructured noise $x_T \sim q_{\text{noise}}$ and perform ancestral sampling $x_{t-1} \sim q(\mathbf{x}_{t-1}|\mathbf{x}_t)$ to obtain new draws from $p_{\text{data}}(\mathbf{x}_0)$. Unfortunately, the backward transition $q(\mathbf{x}_{t-1}|\mathbf{x}_t)$ is mostly intractable due to the marginalization over the entire data distribution. Therefore, we resort to approximating it with a parameterized distribution $p_{\boldsymbol{\theta}}(\mathbf{x}_{t-1}|\mathbf{x}_t)$ at each step $t$. This results in a generative model $p_{\boldsymbol{\theta}}(\mathbf{x}_0, \mathbf{x}_1, \ldots, \mathbf{x}_T) = p_{\boldsymbol{\theta}}(\mathbf{x}_T)\prod_{t=1}^{T} p_{\boldsymbol{\theta}}(\mathbf{x}_{t-1}|\mathbf{x}_t)$, which can be trained by maximizing the evidence lower bound (ELBO) of $\log p_{\boldsymbol{\theta}}(\mathbf{x}_0)$,

$$\log p_{\boldsymbol{\theta}}(\mathbf{x}_0) \geq \mathcal{L}_1(\boldsymbol{\theta}) - \sum_{t=2}^{T} \mathcal{L}_t(\boldsymbol{\theta}) + \text{const.},$$

with $\mathcal{L}_t(\boldsymbol{\theta}) := \mathbb{E}_{q(\mathbf{x}_t|\mathbf{x}_0)}\left[\text{KL}(q(\mathbf{x}_{t-1}|\mathbf{x}_t, \mathbf{x}_0) \parallel p_{\boldsymbol{\theta}}(\mathbf{x}_{t-1}|\mathbf{x}_t))\right]$. For the case $t = 1$, we have $\mathcal{L}_1(\boldsymbol{\theta}) := \mathbb{E}_{q(\mathbf{x}_1|\mathbf{x}_0)}\left[\log p_{\boldsymbol{\theta}}(\mathbf{x}_0|\mathbf{x}_1)\right]$. The ELBO decomposes into a sum of KL divergences between the *conditional backward transition* $q(\mathbf{x}_{t-1}|\mathbf{x}_t, \mathbf{x}_0)$ and $p_{\boldsymbol{\theta}}(\mathbf{x}_{t-1}|\mathbf{x}_t)$ at each time step $t$. Note that $q(\mathbf{x}_{t-1}|\mathbf{x}_t, \mathbf{x}_0) \propto q(\mathbf{x}_t|\mathbf{x}_{t-1})q(\mathbf{x}_{t-1}|\mathbf{x}_0)$ can be calculated analytically for most discrete diffusion models. To define the distribution $p_{\boldsymbol{\theta}}(\mathbf{x}_{t-1}|\mathbf{x}_t)$, previous work (Hoogeboom et al., 2021) suggests that it can be parameterized in a similar manner to $q(\mathbf{x}_{t-1}|\mathbf{x}_t, \mathbf{x}_0)$ by letting

$p_{\boldsymbol{\theta}}(\mathbf{x}_{t-1}|\mathbf{x}_t) = q(\mathbf{x}_{t-1}|\mathbf{x}_t, f(\mathbf{x}_t; \boldsymbol{\theta}))$, where a neural network $f(\mathbf{x}_t; \boldsymbol{\theta})$ is adopted to predict $\mathbf{x}_0$. Typically $f(\mathbf{x}_t; \boldsymbol{\theta}) \in (0, 1)^K$ is the model output normalized by a softmax function, representing the probability vector for each token. A more detailed review and relevant derivations about discrete diffusion models are provided in Appendix A.

## 3 REPARAMETERIZING BACKWARD PROCESSES

This section presents an in-depth study of discrete diffusion probabilistic models. We derive an alternative formulation for the backward process (§3.1) and devise a reparameterized sampling scheme (§3.2), which paves the way for a more generic family of discrete diffusion models (§4).

### 3.1 AN ALTERNATIVE BACKWARD FORMULATION

We first elaborate on how the conditional backward transition of existing discrete diffusion models can be written in a more compact formulation (see Appendix B for the proof).

**Proposition 3.1.** *Let the forward transition of discrete diffusion be* $q(\mathbf{x}_t|\mathbf{x}_{t-1}) = \beta_t \mathbf{x}_{t-1} + (1 - \beta_t) q_{noise}$. *Then the conditional backward transition* $q(\mathbf{x}_{t-1}|\mathbf{x}_t, \mathbf{x}_0)$ *can be equivalently written as*

$$q(\mathbf{x}_{t-1}|\mathbf{x}_t, \mathbf{x}_0) = \begin{cases} \lambda_{t-1}^{(1)} \mathbf{x}_t + \left(1 - \lambda_{t-1}^{(1)}\right) q_{\text{noise}}, & \text{if } \mathbf{x}_t = \mathbf{x}_0 \\ \lambda_{t-1}^{(2)} \mathbf{x}_0 + \left(1 - \lambda_{t-1}^{(2)}\right) q_{\text{noise}}(\mathbf{x}_t), & \text{if } \mathbf{x}_t \neq \mathbf{x}_0. \end{cases} \quad (2)$$

*Here* $q_{\text{noise}}(\mathbf{x}_t) = \beta_t \mathbf{x}_t + (1 - \beta_t) q_{\text{noise}}$ *denotes a noise distribution that interpolates between* $\mathbf{x}_t$ *and* $q_{\text{noise}}$, $\lambda_{t-1}^{(1)} := 1 - \frac{(1-\beta_t)(1-\alpha_{t-1})q_{\text{noise}}(\mathbf{u}=\mathbf{x}_t)}{\alpha_t + (1-\alpha_t)q_{\text{noise}}(\mathbf{u}=\mathbf{x}_t)}$, *and* $\lambda_{t-1}^{(2)} := \frac{\alpha_{t-1}-\alpha_t}{1-\alpha_t}$, *where* $q_{\text{noise}}(\mathbf{u} = \mathbf{x}_t)$ *is the probability of the noise equal to* $\mathbf{x}_t$.

Intuitively, Equation 2 reveals that the main mechanism of the backward process is to shuttle discrete tokens between fully noisy states and the ground truth state $\mathbf{x}_0$, conditioned on the equality of $\mathbf{x}_t$ and $\mathbf{x}_0$. If $\mathbf{x}_t = \mathbf{x}_0$, the current token state is possibly noise-free, and the model either remains noise-free by copying the state $\mathbf{x}_{t-1} \leftarrow \mathbf{x}_t$, or resets the state to the noise. If $\mathbf{x}_t \neq \mathbf{x}_0$, the current state is considered noisy, and the model opts to denoise $\mathbf{x}_t$ to $\mathbf{x}_0$ or remains noisy otherwise. The probability of moving noisy tokens to ground truth states or turning denoised tokens back to noise is governed by $\lambda_{t-1}^{(2)}$ and $1 - \lambda_{t-1}^{(1)}$, respectively.

### 3.2 REPARAMETERIZED SAMPLING

Next, we demonstrate that sampling from the backward transition can be conducted via an augmented path, leading to our full reparameterization. We make use of the simple fact that the mixture distribution in Equation 2 can be sampled in two steps: first, randomly select a component according to their weight, and then sample from the corresponding component distribution. Concisely, we have

$$b_t = \mathbf{1}_{\mathbf{x}_t = \mathbf{x}_0}$$

$$v_{t-1}^{(1)} \sim \text{Bernoulli}\left(\lambda_{t-1}^{(1)}\right), \quad \mathbf{u}_t^{(1)} \sim q_{\text{noise}}$$

$$v_{t-1}^{(2)} \sim \text{Bernoulli}\left(\lambda_{t-1}^{(2)}\right), \quad \mathbf{u}_t^{(2)} \sim q_{\text{noise}}(\mathbf{x}_t)$$

$$\mathbf{x}_{t-1} = b_t \left[ v_{t-1}^{(1)} \mathbf{x}_t + \left(1 - v_{t-1}^{(1)}\right) \mathbf{u}_t^{(1)} \right] + (1 - b_t) \left[ v_{t-1}^{(2)} \mathbf{x}_0 + \left(1 - v_{t-1}^{(2)}\right) \mathbf{u}_t^{(2)} \right]. \quad (3)$$

To simplify the notation, we denote $\mathbf{v}_{t-1} := \left[ v_{t-1}^{(1)}, v_{t-1}^{(2)} \right]$ and $\boldsymbol{\lambda}_{t-1} := \left[ \lambda_{t-1}^{(1)}, \lambda_{t-1}^{(2)} \right]$. This reparameterized backward transition highlights an underlying *routing* mechanism, where the model routes tokens to different distributions according to $\mathbf{v}_{t-1}$: given $b_t$ that discriminates which tokens are currently noisy, $v_{t-1}^{(2)}$ selects and denoises noisy tokens to recover the ground truth $\mathbf{x}_0$, while $v_{t-1}^{(1)}$ determines which denoised tokens revert to the noisy state.

## 4 REPARAMETERIZED DISCRETE DIFFUSION MODELS

In this section, we introduce our proposed diffusion models that reflect the reparameterization (§4.1), which imply effective training (§4.2) and sampling (§4.3) algorithms.

## 4.1 JOINT DIFFUSION MODELING

The routing mechanism in §3.2 works in a latent manner; that is, it is only active during the sampling process but marginalized when advancing the distribution of $\mathbf{x}_{t-1}$. To fully utilize the potential of the developed reparameterization, we propose to elevate the latent routing mechanism to the formulation explicitly and model the *joint* $q(\mathbf{x}_{t-1}, \mathbf{v}_{t-1}|\mathbf{x}_t, \mathbf{x}_0) = q(\mathbf{v}_{t-1})q(\mathbf{x}_{t-1}|\mathbf{v}_{t-1}, \mathbf{x}_t, \mathbf{x}_0)$, where

$$q(\mathbf{v}_{t-1}) = \text{Bernoulli}\left(\boldsymbol{\lambda}_{t-1}\right)$$

$$q(\mathbf{x}_{t-1}|\mathbf{v}_{t-1}, \mathbf{x}_t, \mathbf{x}_0) = \begin{cases} v_{t-1}^{(1)}\mathbf{x}_t + \left(1 - v_{t-1}^{(1)}\right) q_{\text{noise}}, & \text{if } b_t = 1 \\ v_{t-1}^{(2)}\mathbf{x}_0 + \left(1 - v_{t-1}^{(2)}\right) q_{\text{noise}}(\mathbf{x}_t), & \text{if } b_t = 0. \end{cases} \tag{4}$$

Note that $b_t = \mathbf{1}_{\mathbf{x}_t=\mathbf{x}_0}$. This can be considered as a standard discrete diffusion model augmented with step-wise routing indicators $\{\mathbf{v}_{t-1}\}_{t=1}^T$ (see Figure 1). It closely relates to the conventional formulation (Equation 2) in that the original backward process amounts to marginalizing out $\mathbf{v}_{t-1}$ at each time step: $q(\mathbf{x}_{t-1}|\mathbf{x}_t, \mathbf{x}_0) = \mathbb{E}_{q(\mathbf{v}_{t-1})}\left[q(\mathbf{x}_{t-1}|\mathbf{v}_{t-1}, \mathbf{x}_t, \mathbf{x}_0)\right]$. Since the distribution over $\mathbf{v}_{t-1}$ is explicitly considered, this joint diffusion model offers improved flexibility and expressiveness. We refer to this class of models as *reparameterized discrete diffusion models* (RDMs), as it yields an equivalent sampling process to the original formulation but via a reparameterized path.

## 4.2 TRAINING

Similar to previous diffusion models (§2), we define a joint generative process $p_{\boldsymbol{\theta}}(\mathbf{x}_{t-1}, \mathbf{v}_{t-1}|\mathbf{x}_t)$ and optimize the ELBO via the following factorization,

$$\log p(\mathbf{x}_0) \geq \mathbb{E}_{q(\mathbf{x}_{1:T}, \mathbf{v}_{1:T}|\mathbf{x}_0)}\left[\log \frac{p_{\boldsymbol{\theta}}(\mathbf{x}_0, \mathbf{x}_{1:T}, \mathbf{v}_{1:T})}{q(\mathbf{x}_{1:T}, \mathbf{v}_{1:T}|\mathbf{x}_0)}\right] := \mathcal{L}_1(\boldsymbol{\theta}) - \sum_{t=2}^T \mathcal{L}_t(\boldsymbol{\theta}) + \text{const.}.$$

Following standard practices in diffusion models (Hoogeboom et al., 2021; Austin et al., 2021), we randomly sample a time step $t$ and optimize $\boldsymbol{\theta}$ with respect to $\mathcal{L}_t(\boldsymbol{\theta})$. In our case, $\mathcal{L}_1(\boldsymbol{\theta}) = \mathbb{E}_{q(\mathbf{x}_1|\mathbf{x}_0)}\left[\log p_{\boldsymbol{\theta}}(\mathbf{x}_0|\mathbf{x}_1)\right]$; for $t > 1$, $\mathcal{L}_t$ decomposes into a sum of two expected KL divergences over $\mathbf{v}_{t-1}$ and $\mathbf{x}_{t-1}$ respectively (see Appendix C for the full derivation),

$$\mathcal{L}_t(\boldsymbol{\theta}) = \mathbb{E}\left[\text{KL}(q(\mathbf{v}_{t-1}) \parallel p_{\boldsymbol{\theta}}(\mathbf{v}_{t-1}))\right] + \mathbb{E}\left[\text{KL}(q(\mathbf{x}_{t-1}|\mathbf{v}_{t-1}, \mathbf{x}_t, \mathbf{x}_0) \parallel p_{\boldsymbol{\theta}}(\mathbf{x}_{t-1}|\mathbf{v}_{t-1}, \mathbf{x}_t))\right], \tag{5}$$

where the expectations are with respect to $q(\mathbf{x}_t|\mathbf{x}_0)$ and $q(\mathbf{x}_t|\mathbf{x}_0)q(\mathbf{v}_{t-1})$, respectively.

**Parameterization.** The decomposition in Equation 5 suggests parameterizing each conditional of $p_{\boldsymbol{\theta}}(\mathbf{x}_{t-1}, \mathbf{v}_{t-1}|\mathbf{x}_t) = p_{\boldsymbol{\theta}}(\mathbf{x}_{t-1}|\mathbf{v}_{t-1}, \mathbf{x}_t)p_{\boldsymbol{\theta}}(\mathbf{v}_{t-1}|\mathbf{x}_t)$ separately. To simplify the model representation, we constrain $p_{\boldsymbol{\theta}}(\mathbf{v}_{t-1}|\mathbf{x}_t)$ to be the same as $q(\mathbf{v}_{t-1})$ so that the first KL term vanishes. In terms of $p_{\boldsymbol{\theta}}(\mathbf{x}_{t-1}|\mathbf{v}_{t-1}, \mathbf{x}_t)$, it needs to approximate $q(\mathbf{x}_{t-1}|\mathbf{v}_{t-1}, \mathbf{x}_t, \mathbf{x}_0)$ for both $\mathbf{x}_0$ and $b_t = \mathbf{1}_{\mathbf{x}_t=\mathbf{x}_0}$. For the former, we approximate $\mathbf{x}_0$ with a neural network output $f(\mathbf{x}_t; \boldsymbol{\theta})$ (§2); for the latter, it is circumvented via a *teacher-forcing* approach. We leverage the fact that $b_t = \mathbf{1}_{\mathbf{x}_t=\mathbf{x}_0}$ is readily available during training and plug the oracle into $p_{\boldsymbol{\theta}}(\mathbf{x}_{t-1}|\mathbf{v}_{t-1}, \mathbf{x}_t)$, which works well empirically and yields interesting implications as presented below.

**Simplified Training Objectives.** So far, we mainly consider the case of diffusing over a single token. We now slightly abuse the term and denote a sequence of tokens at $t$-th time step as $\mathbf{x}_{t,1:N} := \{\mathbf{x}_{t,n}\}_{n=1}^N$, where $\mathbf{x}_{t,n}$ is the $n$-th token and $N$ is the sequence length.[1] We also assume distributions factorize over each token. We show that the training objective $\mathcal{L}_t(\boldsymbol{\theta})$ for sequence $\mathbf{x}_{t,1:N}$ at the $t$-th step can be reduced to a surprisingly simple expression (see Appendix D for the proof and its connection to previous works) as follows,

$$\mathcal{L}_t(\boldsymbol{\theta}) = \mathbb{E}_{p_{\text{data}}(\mathbf{x}_{0,1:N}) \prod_{n=1}^N q(\mathbf{x}_{t,n}|\mathbf{x}_{0,n})}\left[-\lambda_{t-1}^{(2)} \sum_{n=1}^N (1 - b_{t,n})\mathbf{x}_{0,n}^\top \log f\left(\mathbf{x}_{t,n}; \boldsymbol{\theta}\right)\right]. \tag{6}$$

Based on this result, training RDMs is equivalent to optimizing the standard *multi-class cross-entropy* loss function, which is evaluated over noisy tokens and weighted by $\lambda_{t-1}^{(2)} = \mathbb{E}\left[\mathbf{v}_{t-1}^{(2)}\right]$.

---

[1] We use $\mathbf{x}_t$ and $\mathbf{x}_{t,n}$ interchangeably to represent a single token when there is no ambiguity.

This formulation is conceptually simpler than that of the original discrete diffusion, which requires evaluating the KL divergence between two complicated categoricals (Hoogeboom et al., 2021). Besides, this formulation establishes the discrete analog of *reweighting training strategies*, which are recognized as common practices for training continuous-domain diffusion models (Ho et al., 2020; Nichol & Dhariwal, 2021; Vahdat et al., 2021; Karras et al., 2022). In particular, we can adjust the weight $\lambda_{t-1}^{(2)}$ to reweigh the cross-entropy function so that it is more amenable for training.

More importantly, we note that the training loss function can be *invariant* with respect to $q(\mathbf{v}_{t-1})$ up to reweighting, where different $q(\mathbf{v}_{t-1})$ lead to the same shared objective except for the weight $\lambda_{t-1}^{(2)}$. This makes it possible to train the neural network with one amenable distribution of $\mathbf{v}_{t-1}$ but share the trained model for sampling among a broad family of diffusion processes indexed by $q(\mathbf{v}_{t-1})$.

### 4.3 SAMPLING

Sampling from discrete diffusion processes generally starts with a sequence comprising only noisy tokens and proceeds by drawing $\mathbf{x}_{t-1}, \mathbf{v}_{t-1} \sim p_{\boldsymbol{\theta}}(\mathbf{x}_{t-1}, \mathbf{v}_{t-1}|\mathbf{x}_t)$ at each step.

**Generating $\mathbf{v}_{t-1}$.** A naïve approach of generating $\mathbf{v}_{t-1}$ is by drawing $\mathbf{v}_{t-1} \sim \text{Bernoulli}(\boldsymbol{\lambda})$. However, this assigns equal routing probability $\boldsymbol{\lambda}_{t-1}$ to all tokens, which may be sub-optimal for discriminating and routing different tokens to distinct states. Fortunately, thanks to §4.2, the training objective can be seamlessly shared across different routing distributions. This observation motivates us to employ a

---

**Algorithm 1** Training RDMs

**Input:** neural network $f(\cdot; \boldsymbol{\theta})$, data distribution $p_{\text{data}}(\mathbf{x}_{0,1:N})$, and a custom reweighting scalar $\lambda_{t-1}$.
**Output:** model parameters $\boldsymbol{\theta}$.
**repeat**
    Draw $\mathbf{x}_{0,1:N} \sim p_{\text{data}}(\mathbf{x}_{0,1:N})$;
    Draw $t \in \text{Uniform}(\{1, \ldots, T\})$;
    **for** $n = 1, 2, \ldots, N$ **do**
        Draw $\mathbf{x}_{t,n} \sim q(\mathbf{x}_{t,n}|\mathbf{x}_{0,n})$;
        Let $b_{t,n} = \mathbf{1}_{\mathbf{x}_{t,n}=\mathbf{x}_{0,n}}$;
    **end for**
    $\mathcal{L}(\boldsymbol{\theta}) = -\lambda_{t-1}\sum_{n=1}^{N}(1-b_{t,n})\mathbf{x}_{0,n}^{\top}\log f(\mathbf{x}_{t,n}; \boldsymbol{\theta})$;
    Minimize $\mathcal{L}(\boldsymbol{\theta})$ with respect to $\boldsymbol{\theta}$;
**until** converged

---

more discriminative routing mechanism, where only tokens with high confidence from neural networks are denoised (Ghazvininejad et al., 2019; Savinov et al., 2022; Chang et al., 2022). At each step, assume $\mathbf{v}_{t-1,n}^{(1)}$ and $\mathbf{v}_{t-1,n}^{(2)}$ are initialized to 1 and 0, respectively. We feed the noisy sequence $\mathbf{x}_{t,1:N}$ into the neural network and collect the output $f(\mathbf{x}_{t,n}; \boldsymbol{\theta})$ for each token $n$. We obtain token scores $s_{t,n}$ by taking the maximum value of $f(\mathbf{x}_{t,n}; \boldsymbol{\theta}) \in (0,1)^K$, which reflects the model confidence. $\mathbf{v}_{t-1,n}$ is then set to 1 only when $s_{t,n}$ is among the $k$ largest scores. Formally,

$$s_{t,n} := \max_{1 \leq j \leq K} f_j(\mathbf{x}_{t,n}; \boldsymbol{\theta}), \quad \mathcal{P}_{t-1} = \arg\text{topk}_{1 \leq n \leq N}\{s_{t,n}\}_{n=1}^{N}, \quad \mathbf{v}_{t-1,n}^{(i)} = \mathbf{1}_{n \in \mathcal{P}_{t-1}}, \tag{7}$$

where $i = 1$ if $b_{t,n} = 1$ and $i = 2$ otherwise. This strategy is more informative, as the router can compare token scores to determine their states. Although $\mathbf{v}_{t-1}$ is defined as a function of model scores and its explicit probability distribution is possibly intractable, the usage of such adaptive mechanism is well justified by the shared objective functions, since the diffusion model corresponding to this implicit distribution can also be trained effectively with the shared surrogate objective.

**Recursive Computation for $b_t$.** Another challenge during decoding stems from the computation $b_{t,n} = \mathbf{1}_{\mathbf{x}_{t,n}=\mathbf{x}_{0,n}}$, which is intractable since we lack access to the ground truth $\mathbf{x}_{0,n}$ as in training. We demonstrate that this issue can be circumvented by leveraging a recursive computation based on the functionality of $\mathbf{v}_{t-1,n}$. Specifically, starting with $b_{T,n} = 0$, we can compute $b_{t-1,n}$ as follows,

$$b_{t-1,n} = \left(b_{t,n} \wedge v_{t-1,n}^{(1)}\right) \vee v_{t-1,n}^{(2)}. \tag{8}$$

Intuitively, $b_{t,1:N}$ represents a frontier set that stores the denoised tokens up to the previous iteration. At the current iteration, we add new tokens to the set if $v_{t-1,n}^{(2)} = 1$ and remove elements from the set if the corresponding $v_{t-1,n}^{(1)} = 0$. The updated set is then read out to form $b_{t-1,1:N}$.[2] Note that the update does not add extra computational costs. Equipped with these results, we can efficiently execute the sampling algorithm without difficulties.

---

[2]Technically, for certain noise distributions, the logic operation $\wedge$ and $\vee$ should be noisy since they should compensate for the possibility that noises drawn from $q_{\text{noise}}$ can also coincide with $\mathbf{x}_0$; however, for most diffusion processes considered in this work, such probability is so small that it can be safely ignored.

## 4.4 IMPLEMENTATION

Algorithms 1 and 2 list the full pseudo-codes for training and sampling of RDMs, respectively. Note that the loop over the sequence length $N$ is computed in parallel. The developed formulation of RDMs offers great flexibility for both training and sampling. For instance, we can pass a custom $\lambda_{t-1}$ to reweigh the loss for training, similar to continuous diffusion models. Besides, the denoised token states $\widetilde{\mathbf{x}}_0$ during decoding can be obtained in various ways, such as sampling with annealed temperatures or simply taking the $\arg\max$ of $f(\mathbf{x}_{t,n};\boldsymbol{\theta})$. Please refer to Appendix E for the full implementation details.

## 5 EXPERIMENTS

We evaluate our model on various text generation benchmarks. Our implementation is based on `FairSeq` toolkit (Ott et al., 2019), and a detailed experimental setup can be found in Appendix E.

### 5.1 MACHINE TRANSLATION

**Setup.** We conducted machine translation experiments on three standard benchmarks: `IWSLT14 DE-EN` (Cettolo et al., 2014), `WMT14 EN-DE` (Bojar et al., 2014), and `WMT16 EN-RO` (Bojar et al., 2016), consisting of around 160K/7K/7K, 4.0M/3K/3K, and 610K/2K/2K training/validation/testing sentence pairs, respectively. We operate on original data for all translation tasks and do *not* adopt knowledge distillation (Kim & Rush, 2016; Gu et al., 2018) that replaces the target side of training data with outputs generated by a pre-trained autoregressive Transformer.

---

**Algorithm 2** Sampling from RDMs

**Input:** trained network $f(\cdot;\boldsymbol{\theta})$ and temperature $\tau$.
**Output:** generated sample $\mathbf{x}_0$.
**for** $n = 1, 2, \ldots, N$ **do**
    Initialize $\mathbf{x}_{T,n} \sim q_{\text{noise}}$;
    Initialize $b_{T,n} = 0$;
**end for**
**for** $t = T, \ldots, 1$ **do**
    **for** $n = 1, 2, \ldots, N$ **do**
        Draw $\widetilde{\mathbf{x}}_{0,n} \sim \text{Categorical}(f(\mathbf{x}_{t,n};\boldsymbol{\theta})/\tau)$;
        Generate $\mathbf{v}_{t-1,n}$ according to Equation 7;
        **if** $b_{t,n} = 1$ **then**
            Draw $\mathbf{u}_{t,n}^{(1)} \sim q_{\text{noise}}$;
            $\mathbf{x}_{t-1,n} = v_{t-1,n}^{(1)}\mathbf{x}_{t,n} + \left(1 - v_{t-1,n}^{(1)}\right)\mathbf{u}_{t,n}^{(1)}$;
        **else**
            Draw $\mathbf{u}_{t,n}^{(2)} \sim q_{\text{noise}}(\mathbf{x}_{t,n})$;
            $\mathbf{x}_{t-1,n} = v_{t-1,n}^{(2)}\widetilde{\mathbf{x}}_{0,n} + \left(1 - v_{t-1,n}^{(2)}\right)\mathbf{u}_{t,n}^{(2)}$;
        **end if**
        Let $b_{t-1,n} = b_{t,n} \wedge v_{t-1,n}^{(1)} \vee v_{t-1,n}^{(2)}$;
    **end for**
**end for**
**Return** $\mathbf{x}_{0,1:N}$.

---

**Results.** As seen from Table 1, existing discrete diffusion models usually perform badly on the translation task and do not scale well for large-sized datasets. In particular, multinomial diffusion achieves worse translation quality albeit with more iterations and fails to decode decently on `WMT14 EN-DE` dataset. The proposed reparameterization yields significant performance boosts (about 3~20 BLEU improvements) on both absorbing and multinomial diffusion across all datasets and iteration steps. We also observe that the performance gain is much more significant for multinomial diffusion, which is possibly due to the fix of its degenerated behavior (more details are in §5.4). Our approach effectively scales diffusion models to larger datasets, outperforming previous non-autoregressive baselines `CMLM` (Ghazvininejad et al., 2019), and achieves promising results even competitive with autoregressive baselines. Besides, RDMs perform significantly better than continuous diffusion models `CDCD` (Dieleman et al., 2022) while running with more than $8\times$ fewer iterations.

### 5.2 QUESTION GENERATION AND PARAPHRASING

**Setup.** We also evaluate the performance of our model on general sequence-to-sequence generation tasks, following Gong et al. (2022). Due to the limited computation budget, the tasks we focus on are 1) **question generation** (`QG`) using the Quasar-T dataset (Dhingra et al., 2017) with approximately 117K/2K/10K pairs for training/validation/testing, and 2) **paraphrasing** with Quora Question Pairs (`QQP`) containing around 145K/2K/2.5K training/validation/testing question pairs.

**Results.** We report comparisons among discrete diffusion, continuous diffusion, and auto-regressive baselines in Table 3. We observe that either variant of RDMs not only improves their vanilla counterparts by a large margin but also outperforms the strong continuous diffusion baseline DiffuSeq as well as various auto-regressive baselines across several evaluation metrics. Besides, DiffuSeq needs 2000 steps to finish the decoding, while RDMs can produce higher-quality samples with 10 iterations, reducing runtime by over $200\times$. This implies that RDMs can attain a much better trade-off between generation quality and runtime. In addition, we also investigate the effect of candidate

Table 1: BLEU score comparisons on `IWSLT14 DE-EN`, `WMT14 EN-DE`, and `WMT16 EN-RO` benchmarks. [*] denotes results reported from previous work.

| | Model | # Iterations | IWSLT14 DE-EN Vanilla | Reparam. | WMT16 EN-RO Vanilla | Reparam. | WMT14 EN-DE Vanilla | Reparam. |
|---|---|---|---|---|---|---|---|---|
| **Continuous Diffusion** | CDCD (Dieleman et al., 2022) | 200 | – | | – | | 20.0[*] | |
| **Discrete Diffusion** | Multinomial Diffusion (Hoogeboom et al., 2021) | 2 | 23.05 | 28.01 | 26.61 | 30.16 | 4.28 | 21.43 |
| | | 4 | 24.24 | 30.57 | 27.81 | 31.70 | 4.31 | 24.05 |
| | | 10 | 21.28 | 32.23 | 25.25 | 33.00 | 6.94 | 25.63 |
| | | 16 | 20.59 | 32.58 | 24.36 | 33.11 | 6.07 | 25.64 |
| | | 25 | 20.06 | 32.84 | 23.94 | 33.31 | 3.69 | 26.04 |
| | Absorbing Diffusion (Austin et al., 2021) | 2 | 25.24 | 27.60 | 27.24 | 30.72 | 16.46 | 21.00 |
| | | 4 | 26.93 | 31.47 | 29.16 | 32.60 | 19.48 | 24.26 |
| | | 10 | 28.32 | 33.91 | 30.41 | 33.38 | 21.62 | 26.96 |
| | | 16 | 28.38 | 34.41 | 30.79 | 33.82 | 22.07 | 27.58 |
| | | 25 | 28.93 | 34.49 | 30.56 | 33.99 | 22.52 | **27.59** |
| Non-autoregressive Models | CMLM (Ghazvininejad et al., 2019) | 16 | 32.18 | | 32.90 | | 25.00 | |
| | CMLM+SMART (Ghazvininejad et al., 2020) | 10 | 30.74[*] | | 32.71[*] | | 25.10[*] | |
| | Levenshtein Transformer (Gu et al., 2019) | Adaptive | - | | - | | 25.20[*] | |
| | DisCo (Kasai et al., 2020) | Adaptive | - | | - | | 25.64[*] | |
| | CMLMC (Huang et al., 2022) | 10 | 34.28[*] | | 34.14[*] | | 26.40[*] | |
| **Autoregressive Models** | Transformer-base (Vaswani et al., 2017) | n.a. | **34.51** | | **34.16** | | 27.53 | |

sample size in Table 8. We notice that DiffuSeq benefits more from large sample sets (e.g., when the sample size increases from 1 to 10) than RDMs. We attribute this to the possibility that adding Gaussian noise to token embeddings in DiffuSeq might lead to more diverse samples. This helps make better use of the MBR decoding, indicating that there might be room to improve RDMs to leverage multiple decodes. Nevertheless, RDMs still achieve better performance than DiffuSeq across both cases of single and multiple samples.

## 5.3 ANALYSIS

**On the Effect of Training and Decoding Strategies.** This section explores the impact of improved training and decoding techniques developed for RDMs. Concerning the *training* aspect, we compare the derived loss objective, which takes the form of a reweighting cross-entropy function (§4.2), with that of traditional discrete diffusion models. As for *decoding*, we evaluate the effectiveness of the discriminative routing mechanism (§4.3) against the vanilla strategy that randomly denoises tokens. The overall comparison is presented in Figure 2a (as well as Figures 3a and 3b in

Table 2: BLEU scores on `IWSLT14 DE-EN` with different reweighting schemes, evaluated under RDM-absorbing with 10 *vanilla/improved decoding* steps.

| Reweighting training scheme | Vanilla | Improved |
|---|---|---|
| Original $\left(\lambda_{t-1}^{(2)} = \frac{\alpha_{t-1} - \alpha_t}{1 - \alpha_t}\right)$ | 28.32 | 30.64 |
| Linear $\left(\lambda_{t-1}^{(2)} = 1 - \frac{t-1}{T}\right)$ | **31.04** | **33.91** |
| Constant $\left(\lambda_{t-1}^{(2)} = 1\right)$ | 29.75 | 32.57 |

Appendix G.4). The results indicate that adopting either the improved training or decoding scheme leads to a significant performance boost over vanilla baselines, which can be further amplified by integrating both. In particular, we discover that the inferior performance of vanilla multinomial diffusion stems from *both* inadequate training and ineffective decoding. Improving either the training or decoding process results in gains of over 10~20 BLEU points, allowing the model to scale well for more decoding steps. Besides, we also ablate the design choice of reweighting strategies (§4.2) in Table 2, and observe that a linear heuristic proposed in (Bond-Taylor et al., 2022) yields large improvements in performance on `IWSLT14 DE-EN`. Therefore, this reweighting scheme is adopted by default unless specified otherwise.

**On Decoding Speed.** This section visualizes the performance-runtime comparison among different text generation models. All models considered here have roughly the same parameter counts (90~110M), and the speed is measured under the setting of 32 batch size on one NVIDIA GeForce RTX 3090 GPU, averaged by 30 runs. As shown in Figure 2b, there is a clear trend that RDMs usually run up to $10\times$ faster than a similar-sized auto-regressive baseline GPT2 (Radford et al., 2019),

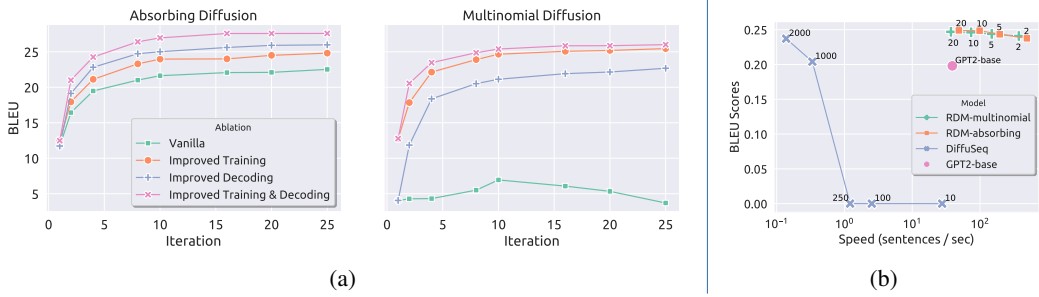

(a)                                                  (b)

Figure 2: **Left**: ablation study of improved training and decoding strategies on `WMT14 EN-DE` test set for absorbing and multinomial diffusion, respectively. **Right**: The quality-speed comparison among different models for `QQP` dataset. The number annotation indicates the iteration *steps* except for GPT2, which does not have a fixed number of iterations. The horizontal axis is in log scale.

while to achieve similar performance, continuous diffusion models are several orders of magnitude slower than GPT2. Furthermore, discrete diffusion models here are trained with 50 time steps in total but are able to achieve satisfactory quality even with 2 or 5 steps. In contrast, the continuous diffusion model DiffuSeq, trained with 2000 time steps, mostly generates non-meaningful sentences (BLEU scores getting close to zero) under down-sampled time steps, even equipped with advanced samplers like DDIM (Song et al., 2021a). This reveals the inherent difference between discrete and continuous diffusion, where discrete diffusion generalizes much better to various setups of iteration steps.

### 5.4 EXAMPLES

In this section, we conduct a qualitative analysis to showcase the advantages of our model by illustrating the generated samples from the `QQP` task. We focus on the case of multinominal diffusion and its reparameterized variant; a more comprehensive analysis can be found in Appendix G.5.

**Multinomial Diffusion Does Not Decode Iteratively.** As seen in Table 4, we observe that across all text generation tasks, vanilla multinomial diffusion only generates the hypothesis at the *first* iteration and gets stuck in the same state thereafter. This means multinomial diffusion decodes in one shot and does not leverage the iterative process for further refinement. In Appendix G.5, we show that this abnormal behavior is primarily due to its degenerated backward formulation, which can be neatly fixed by our reparameterization. The resulting behavior is much more expected and leads to better generation quality.

**The Slow Convergence of Continuous Diffusion.** We also demonstrate the down-sampled dynamics during the generation of DiffuSeq. In contrast to discrete diffusion models, where relevant tokens emerge within only a few steps, continuous diffusion hardly decodes meaningful tokens until the 1000-th iteration or later. This validates our hypothesis that the Gaussian diffusion over token embeddings is noisy and slow by design; furthermore, many diffusing steps are required to emit a significant change over token states due to the rounding quantization (see Table 12 for an illustration).

### 6 RELATED WORK

#### 6.1 DIFFUSION MODELS FOR TEXT GENERATION

**Text Generation with Discrete Diffusion.** Discrete diffusion processes have close connections with previously developed language models. For instance, traditional auto-regressive language models can be seen as a special deterministic discrete diffusion process (Austin et al., 2021). In addition, D3PMs (Austin et al., 2021) also introduce an absorbing diffusion strongly linked to masked language models (Devlin et al., 2019). The diffusion formulation is further generalized in various aspects, such as enabling editing-based operations (Johnson et al., 2021) or casting generic permuted language models (Yang et al., 2019) as a diffusion process (Hoogeboom et al., 2022a).

Table 3: Comparisons among different text generators on `QG` and `QQP` tasks. [†] numbers are taken from Gong et al. (2022). All discrete diffusion models are run with 10 iterations.

| Task | Model | BLEU ↑ | ROUGE-L ↑ | BERTScore ↑ | Dist-1↑ |
|------|-------|--------|-----------|-------------|---------|
| QG | Transformer-base[†] | 0.1663 | 0.3441 | 0.6307 | 0.9309 |
| | GPT2-base FT[†] | 0.0741 | 0.2714 | 0.6052 | 0.9602 |
| | GPT2-large FT[†] | 0.1110 | 0.3215 | 0.6346 | **0.9670** |
| | GPVAE-T5[†] | 0.1251 | 0.3390 | 0.6308 | 0.9381 |
| | NAR-LevT[†] | 0.0930 | 0.2893 | 0.5491 | 0.8914 |
| | DiffuSeq[†] | 0.1731 | **0.3665** | 0.6123 | 0.9056 |
| | Absorbing | 0.1738 | 0.3503 | 0.6312 | 0.9095 |
| | RDM-absorbing | 0.1791 | 0.3565 | **0.6393** | 0.9202 |
| | Multinomial | 0.1696 | 0.3429 | 0.6188 | 0.8990 |
| | RDM-multinomial | **0.1802** | 0.3550 | 0.6310 | 0.9082 |
| QQP | Transformer-base[†] | **0.2722** | 0.5748 | 0.8381 | 0.9748 |
| | GPT2-base FT[†] | 0.1980 | 0.5212 | 0.8246 | 0.9798 |
| | GPT2-large FT[†] | 0.2059 | 0.5415 | 0.8363 | 0.9819 |
| | GPVAE-T5[†] | 0.2409 | 0.5886 | 0.8466 | 0.9688 |
| | NAR-LevT[†] | 0.2268 | 0.5795 | 0.8344 | 0.9790 |
| | DiffuSeq[†] | 0.2413 | 0.5880 | 0.8365 | 0.9807 |
| | Absorbing | 0.2382 | 0.5834 | 0.8294 | 0.9566 |
| | RDM-absorbing | 0.2510 | **0.5945** | **0.8472** | **0.9849** |
| | Multinomial | 0.2070 | 0.5539 | 0.7985 | 0.9175 |
| | RDM-multinomial | 0.2498 | 0.5886 | 0.8466 | 0.9817 |

Table 4: Qualitative samples of test paraphrases generated from different diffusion models on `QQP` dataset. [‡] texts are truncated to fit into the table. Words are in lower case.

**Source:** how can one increase concentration?
**Reference:** how can i improve my concentration?

| | # Iter. | Decodes |
|---|---------|---------|
| DiffuSeq | 0 | ∘ skeptical coli ##zam gael erika calves wharf [unused791][‡] |
| | 500 | ∘ cessna i perez newark ? venezuelan regeneration 283 zhejiang[‡] |
| | 1000 | ∘ johanna 730 i improve terminals ? |
| | 1500 | ∘ how do i improve concentration ? |
| | 2000 | ∘ how do i improve concentration ? |
| Multinomial | 0 | ∘ ##tly distances outline ##cera khmer curvature question ##tl |
| | 1 | ∘ how can i improve focus in concentration ? |
| | 2 | ∘ how can i improve focus in concentration ? |
| | 3 | ∘ how can i improve focus in concentration ? |
| | 4 | ∘ how can i improve focus in concentration ? |
| | 5 | ∘ how can i improve focus in concentration ? |
| RDM-multinomial | 0 | ∘ lungs ##down intensity cortes ##lden ufo oldies |
| | 1 | ∘ worker blurted i ##kal caledonia concentration ##vb |
| | 2 | ∘ how trait i ##kal my concentration ##vb |
| | 3 | ∘ how trait i increase my concentration ? |
| | 4 | ∘ how trait i increase my concentration ? |
| | 5 | ∘ how do i increase my concentration ? |

The text generation performance of discrete diffusion processes is initially evaluated on language modeling tasks (Hoogeboom et al., 2021; Austin et al., 2021), despite limited success. Recent studies have improved the performance of discrete diffusion on various tasks by devising unrolling training strategies (Savinov et al., 2022), combining pre-trained models (He et al., 2022), incorporating autoregressive decoding with editing-based refinements (Reid et al., 2022), or leveraging a more effective diffusion process over data modalities with advanced search algorithms (Qian et al., 2022).

**Text Generation with Continuous Diffusion.** There has been a surge of recent interest in adapting continuous diffusion models for text generation. This approach typically applies Gaussian diffusion over the embedding space, achieving impressive results (Li et al., 2022b; Gong et al., 2022; Dieleman et al., 2022; Strudel et al., 2022; Lin et al., 2022; Yuan et al., 2022; Gao et al., 2022; Ye et al., 2023). Some studies convert discrete tokens to bit strings and model them as real values (Chen et al., 2022), or inject Gaussian noise into token logits instead of embeddings (Han et al., 2022). Other studies focus on learning a latent continuous diffusion of pre-trained auto-regressive models (Wang et al., 2022; Lovelace et al., 2022). For a detailed review of recent advances in diffusion models, we refer readers to Cao et al. (2022); Croitoru et al. (2022); Yang et al. (2022); Li et al. (2023).

## 6.2 ITERATIVE NON-AUTOREGRESSIVE TEXT GENERATION

Diffusion-based generative models are closely related to iterative non-autoregressive generation (Gu et al., 2018) in the context of machine translation. The generation process often involves iterative refinement (Lee et al., 2018; Ghazvininejad et al., 2019; Stern et al., 2019; Gu et al., 2019; Kasai et al., 2020; Ghazvininejad et al., 2020; Huang et al., 2022). Our adaptive routing mechanism (§4.3) takes inspiration from the heuristic used in CMLM (Ghazvininejad et al., 2019), which refines the sequence by masking tokens with low model confidence. However, unlike CMLM, our approach only integrates the masking heuristic within the routing mechanism at each diffusion step, rather than relying entirely on it for decoding. This makes the decoding procedure governed by the formulated diffusion process and helps achieve better performance in practice.

## 7 CONCLUSION

This work presents an extensive analysis of discrete diffusion models. Based on the developed understanding, we propose a family of *reparameterized discrete diffusion models* (RDMs) that significantly improve previous work in both training and decoding. We evaluate the proposed model family in various text generation benchmarks and demonstrate the boosted generation quality.

RDMs define a general framework for discrete diffusion processes and can be extended in several ways. For instance, RDMs are currently confined to generating *fixed-length* sentences and rely on an explicit length prediction module to propose the sequence length. It would be interesting to extend

the model to enable variable-length sequence generation. Besides, our proposed adaptive routing mechanism (§4.3) makes the initial attempt to unleash the expressiveness of RDMs; the shared training objective (§4.2) allows more advanced search methods to be incorporated into the sampling process for better generation quality. A further investigation into this direction is left as future work.

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

# Appendices

## A    EXTENDED BACKGROUND ABOUT DISCRETE DIFFUSION MODELS

Discrete diffusion probabilistic models are first explored in Sohl-Dickstein et al. (2015) for Bernoulli data. Multinomial diffusion (Hoogeboom et al., 2021) later proposes a uniform corruption process for categorical variables, which are extended by D3PMs (Austin et al., 2021) to support general transition matrices, including an absorbing variant that draws close connections to masked language models (Devlin et al., 2019). Several recent works push this line of research further in various aspects, such as incorporating editing-based operations (Johnson et al., 2021; Reid et al., 2022), casting permuted language models (Yang et al., 2019) as diffusion models (Hoogeboom et al., 2022a), developing a continuous-time framework (Campbell et al., 2022), as well as exploring an analog of score functions for learning the reverse process Sun et al. (2022).

**Applications.**    Discrete diffusion has been applied to a variety of tasks, including graph generation (Seff et al., 2019; Haefeli et al., 2022; Vignac et al., 2022), image generation (Esser et al., 2021; Bond-Taylor et al., 2022; Gu et al., 2022; Tang et al., 2022; Hu et al., 2022a), vision-language generation (Hu et al., 2022b), and general multimodal conditional synthesis (Zhu et al., 2022). Lezama et al. (2022) draws connections between discrete diffusion processes and non-autoregressive Transformers for visual domains (Chang et al., 2022; Yu et al., 2022; Chang et al., 2023). A detailed discussion about applications to natural language texts can be found at §6.

### A.1    THE DERIVATION OF ELBO

Discrete diffusion models are typically trained by maximizing a lower bound of its marginal log-likelihood, defined below,

$$
\begin{aligned}
&\log p_{\boldsymbol{\theta}}(\mathbf{x}_0) \\
&= \log \int p_{\boldsymbol{\theta}}(\mathbf{x}_0, \mathbf{x}_1, \ldots, \mathbf{x}_T) d\mathbf{x}_1 \cdots d\mathbf{x}_T \\
&= \log \int \frac{p_{\boldsymbol{\theta}}(\mathbf{x}_0, \mathbf{x}_1, \ldots, \mathbf{x}_T)}{q(\mathbf{x}_1, \ldots, \mathbf{x}_T | \mathbf{x}_0)} q(\mathbf{x}_1, \ldots, \mathbf{x}_T | \mathbf{x}_0) d\mathbf{x}_1 \cdots d\mathbf{x}_T \\
&= \log \mathbb{E}_{q(\mathbf{x}_1, \ldots, \mathbf{x}_T | \mathbf{x}_0)} \left[ \frac{p_{\boldsymbol{\theta}}(\mathbf{x}_0, \mathbf{x}_1, \ldots, \mathbf{x}_T)}{q(\mathbf{x}_1, \ldots, \mathbf{x}_T | \mathbf{x}_0)} \right] \\
&\geq \mathbb{E}_{q(\mathbf{x}_1, \ldots, \mathbf{x}_T | \mathbf{x}_0)} \left[ \log \frac{p_{\boldsymbol{\theta}}(\mathbf{x}_0, \mathbf{x}_1, \ldots, \mathbf{x}_T)}{q(\mathbf{x}_1, \ldots, \mathbf{x}_T | \mathbf{x}_0)} \right] \\
&= \mathbb{E}_{q(\mathbf{x}_1, \ldots, \mathbf{x}_T | \mathbf{x}_0)} \left[ \log \frac{p_{\boldsymbol{\theta}}(\mathbf{x}_0 | \mathbf{x}_1) p_{\boldsymbol{\theta}}(\mathbf{x}_T) \prod_{t=2}^{T} p_{\boldsymbol{\theta}}(\mathbf{x}_{t-1} | \mathbf{x}_t)}{q(\mathbf{x}_T | \mathbf{x}_0) \prod_{t=2}^{T} q(\mathbf{x}_{t-1} | \mathbf{x}_t, \mathbf{x}_0)} \right] \\
&= \mathbb{E}_{q(\mathbf{x}_1, \ldots, \mathbf{x}_T | \mathbf{x}_0)} \left[ \log p_{\boldsymbol{\theta}}(\mathbf{x}_0 | \mathbf{x}_1) - \sum_{t=2}^{T} \log \frac{q(\mathbf{x}_{t-1} | \mathbf{x}_t, \mathbf{x}_0)}{p_{\boldsymbol{\theta}}(\mathbf{x}_{t-1} | \mathbf{x}_t)} - \log \frac{q(\mathbf{x}_T | \mathbf{x}_0)}{p_{\boldsymbol{\theta}}(\mathbf{x}_T)} \right] \\
&= \mathbb{E}_q \left[ \log p_{\boldsymbol{\theta}}(\mathbf{x}_0 | \mathbf{x}_1) - \sum_{t=2}^{T} \mathrm{KL}(q(\mathbf{x}_{t-1} | \mathbf{x}_t, \mathbf{x}_0) \,\|\, p_{\boldsymbol{\theta}}(\mathbf{x}_{t-1} | \mathbf{x}_t)) - \mathrm{KL}(q(\mathbf{x}_T | \mathbf{x}_0) \,\|\, p_{\boldsymbol{\theta}}(\mathbf{x}_T)) \right] \\
&= \underbrace{\mathbb{E}_{q(\mathbf{x}_1 | \mathbf{x}_0)} \left[ \log p_{\boldsymbol{\theta}}(\mathbf{x}_0 | \mathbf{x}_1) \right]}_{\mathcal{L}_1(\boldsymbol{\theta})} - \sum_{t=2}^{T} \underbrace{\mathbb{E}_{q(\mathbf{x}_t | \mathbf{x}_0)} \left[ \mathrm{KL}(q(\mathbf{x}_{t-1} | \mathbf{x}_t, \mathbf{x}_0) \,\|\, p_{\boldsymbol{\theta}}(\mathbf{x}_{t-1} | \mathbf{x}_t)) \right]}_{\mathcal{L}_t(\boldsymbol{\theta})} + \mathrm{const.} \quad (9)
\end{aligned}
$$

### A.2    PARAMETERIZATION

Recall that our objective is to minimize the KL divergence between $q(\mathbf{x}_{t-1} | \mathbf{x}_t, \mathbf{x}_0)$ and a parameterized distribution $p_{\boldsymbol{\theta}}(\mathbf{x}_{t-1} | \mathbf{x}_t)$ at each time step. A widely adopted way is then defining $p_{\boldsymbol{\theta}}(\mathbf{x}_{t-1} | \mathbf{x}_t) = q(\mathbf{x}_{t-1} | \mathbf{x}_t, \tilde{\mathbf{x}}_0)$, where $\tilde{\mathbf{x}}_0 = f(\mathbf{x}_t; \boldsymbol{\theta})$ is predicted by a Transformer model. Austin et al. (2021)

considers an alternative parameterization by letting $p_{\boldsymbol{\theta}}(\mathbf{x}_{t-1}|\mathbf{x}_t) \propto \sum_{\widetilde{\mathbf{x}}_0} q(\mathbf{x}_{t-1}, \mathbf{x}_t|\widetilde{\mathbf{x}}_0)p_{\boldsymbol{\theta}}(\widetilde{\mathbf{x}}_0|\mathbf{x}_t)$, where we learn $p_{\boldsymbol{\theta}}(\widetilde{\mathbf{x}}_0|\mathbf{x}_t)$ similarly to $f(\mathbf{x}_t; \boldsymbol{\theta})$. These two approaches are different in general and define distinct generative processes in general; we follow the former method due to its simplicity and conciseness. More details can be found below.

### A.3  BACKWARD TRANSITION PROBABILITIES

This section provides separate derivations for the original backward transition formulation of various discrete diffusion processes.

**Absorbing Diffusion.**  The absorbing diffusion (Austin et al., 2021) defines a Markov chain where a token goes into an absorbing mask state denoted by $[M]$ with some probability at each time step and stays the same thereafter. The forward transition probability is defined as $q(\mathbf{x}_t|\mathbf{x}_{t-1}) = \beta_t \mathbf{x}_{t-1} + (1 - \beta_t)q_{\text{noise}}$, where $q_{\text{noise}} = [M]$ is the point mass with all of the probability on an absorbing state (the mask state [M] is denoted as a one-hot vector).

Regarding the conditional backward transition probability $q(\mathbf{x}_{t-1}|\mathbf{x}_t, \mathbf{x}_0)$, $\mathbf{x}_t$ can only stay in either state $\mathbf{x}_0$ or state $[M]$. If $\mathbf{x}_t = \mathbf{x}_0$, then $\mathbf{x}_{t-1}$ must also be in state $\mathbf{x}_0$ since it is not absorbed yet; while if $\mathbf{x}_t = \mathbf{e}_{[M]}$, we have

$$q(\mathbf{x}_{t-1} = [M]|\mathbf{x}_t = [M], \mathbf{x}_0) = \frac{q(\mathbf{x}_t = [M]|\mathbf{x}_{t-1} = [M])q(\mathbf{x}_{t-1} = [M]|\mathbf{x}_0)}{q(\mathbf{x}_t = [M]|\mathbf{x}_0)}$$

$$= \frac{1 \cdot (1 - \alpha_{t-1})}{1 - \alpha_t} = \frac{1 - \alpha_{t-1}}{1 - \alpha_t};$$

$$q(\mathbf{x}_{t-1} = \mathbf{x}_0|\mathbf{x}_t = [M], \mathbf{x}_0) = \frac{q(\mathbf{x}_t = [M]|\mathbf{x}_{t-1} = \mathbf{x}_0)q(\mathbf{x}_{t-1} = \mathbf{x}_0|\mathbf{x}_0)}{q(\mathbf{x}_t = [M]|\mathbf{x}_0)}$$

$$= \frac{(1 - \beta_t)\alpha_{t-1}}{1 - \alpha_t} = \frac{\alpha_{t-1} - \alpha_t}{1 - \alpha_t}.$$

The actual generative process is defined as $p_{\boldsymbol{\theta}}(\mathbf{x}_{t-1}|\mathbf{x}_t) \propto \sum_{\widetilde{\mathbf{x}}_0} q(\mathbf{x}_{t-1}, \mathbf{x}_t|\widetilde{\mathbf{x}}_0)p_{\boldsymbol{\theta}}(\widetilde{\mathbf{x}}_0|\mathbf{x}_t)$, where we predict the probability vector $p_{\boldsymbol{\theta}}(\widetilde{\mathbf{x}}_0|\mathbf{x}_t) := f_{\widetilde{\mathbf{x}}_0}(\mathbf{x}_t; \boldsymbol{\theta})$ from a Transformer. As shown in Austin et al. (2021), this formulation has a simple expression. Suppose $k \neq [M]$ is one of the $K$ possible states. Note that

- if $\mathbf{x}_t = k \neq [M]$, then due to the joint $q(\mathbf{x}_{t-1}, \mathbf{x}_t|\widetilde{\mathbf{x}}_0)$ there is only one entry that is non-zero within the sum $q(\mathbf{x}_{t-1} = k, \mathbf{x}_t = k|\widetilde{\mathbf{x}}_0 = k)p_{\boldsymbol{\theta}}(\widetilde{\mathbf{x}}_0 = k|\mathbf{x}_t)$. As a result, the reverse distribution becomes a point mass over position $k$;

- if $\mathbf{x}_t = [M]$, we have

$$p_{\boldsymbol{\theta}}(\mathbf{x}_{t-1} = [M]|\mathbf{x}_t = [M]) \propto \sum_{\widetilde{\mathbf{x}}_0} q(\mathbf{x}_t = [M]|\mathbf{x}_{t-1} = [M])q(\mathbf{x}_{t-1} = [M]|\widetilde{\mathbf{x}}_0)p_{\boldsymbol{\theta}}(\widetilde{\mathbf{x}}_0|\mathbf{x}_t)$$

$$= \sum_{\widetilde{\mathbf{x}}_0} (1 - \alpha_{t-1})p_{\boldsymbol{\theta}}(\widetilde{\mathbf{x}}_0|\mathbf{x}_t)$$

$$= (1 - \alpha_{t-1}) \sum_{\widetilde{\mathbf{x}}_0} p_{\boldsymbol{\theta}}(\widetilde{\mathbf{x}}_0|\mathbf{x}_t)$$

$$= 1 - \alpha_{t-1};$$

$$p_{\boldsymbol{\theta}}(\mathbf{x}_{t-1} = k|\mathbf{x}_t = [M]) \propto \sum_{\widetilde{\mathbf{x}}_0} q(\mathbf{x}_t = [M]|\mathbf{x}_{t-1} = k)q(\mathbf{x}_{t-1} = k|\widetilde{\mathbf{x}}_0)p_{\boldsymbol{\theta}}(\widetilde{\mathbf{x}}_0|\mathbf{x}_t)$$

$$= q(\mathbf{x}_t = [M]|\mathbf{x}_{t-1} = k)q(\mathbf{x}_{t-1} = k|\widetilde{\mathbf{x}}_0 = k)p_{\boldsymbol{\theta}}(\widetilde{\mathbf{x}}_0 = k|\mathbf{x}_t)$$

$$= (\alpha_{t-1} - \alpha_t)p_{\boldsymbol{\theta}}(\widetilde{\mathbf{x}}_0 = k|\mathbf{x}_t).$$

They can be easily normalized as well,

$$p_{\boldsymbol{\theta}}(\mathbf{x}_{t-1} = [M]|\mathbf{x}_t = [M]) = \frac{1 - \alpha_{t-1}}{1 - \alpha_t}$$

$$p_{\boldsymbol{\theta}}(\mathbf{x}_{t-1} = k|\mathbf{x}_t = [M]) = \frac{(\alpha_{t-1} - \alpha_t)p_{\boldsymbol{\theta}}(\widetilde{\mathbf{x}}_0 = k|\mathbf{x}_t)}{1 - \alpha_t}.$$

This can be written in a more compact way, where $p_{\boldsymbol{\theta}}(\widetilde{\mathbf{x}}_0 = k|\mathbf{x}_t) := f_k(\mathbf{x}_t; \boldsymbol{\theta})$ and $q_{\text{noise}} = [M]$ is a point mass with all the probability put over the absorbing state $[M]$.

$$p_{\boldsymbol{\theta}}(\mathbf{x}_{t-1}|\mathbf{x}_t) = \begin{cases} \left(1 - \frac{1-\alpha_{t-1}}{1-\alpha_t}\right) f(\mathbf{x}_t; \boldsymbol{\theta}) + \frac{1-\alpha_{t-1}}{1-\alpha_t} q_{\text{noise}}, & \text{if } \mathbf{x}_t = [M] \\ \mathbf{x}_t, & \text{if } \mathbf{x}_t \neq [M]. \end{cases} \tag{10}$$

We can also generalize this result for $0 < s < t \leq T$. For the forward transition, we have $q(\mathbf{x}_t|\mathbf{x}_s = \mathbf{x}_0) = \prod_{i=s+1}^{t} \beta_i \mathbf{x}_0 + \left(1 - \prod_{i=s+1}^{t} \beta_i\right) q_{\text{noise}} = \frac{\alpha_t}{\alpha_s} \mathbf{x}_0 + \frac{\alpha_s - \alpha_t}{\alpha_s} q_{\text{noise}}$. Thus the backward transition can be derived as well according to Bayes' rule,

$$q(\mathbf{x}_s|\mathbf{x}_t, \mathbf{x}_0) = \begin{cases} \frac{\alpha_s - \alpha_t}{1-\alpha_t} \mathbf{x}_0 + \frac{1-\alpha_s}{1-\alpha_t} q_{\text{noise}}, & \text{if } \mathbf{x}_t = [M] \\ \mathbf{x}_t, & \text{if } \mathbf{x}_t \neq [M]. \end{cases} \tag{11}$$

$$p_{\boldsymbol{\theta}}(\mathbf{x}_s|\mathbf{x}_t) = \begin{cases} \frac{\alpha_s - \alpha_t}{1-\alpha_t} f(\mathbf{x}_t; \boldsymbol{\theta}) + \frac{1-\alpha_s}{1-\alpha_t} q_{\text{noise}}, & \text{if } \mathbf{x}_t = [M] \\ \mathbf{x}_t, & \text{if } \mathbf{x}_t \neq [M]. \end{cases} \tag{12}$$

**Multinomial Diffusion.** In multinomial diffusion (Hoogeboom et al., 2021), the forward transition probability is defined as $q(\mathbf{x}_t|\mathbf{x}_{t-1}) = \beta_t \mathbf{x}_{t-1} + (1 - \beta_t) q_{\text{noise}}$, where $q_{\text{noise}} = \mathbf{1}/K$ is a uniform distribution over $\{1, 2, \ldots, K\}$ with $\mathbf{1}$ is a $K$-dimensional vector with all ones. The backward transition probability conditional on the original data $\mathbf{x}_0$ can be derived according to Bayes' rule and in closed form:

$$q(\mathbf{x}_{t-1}|\mathbf{x}_t, \mathbf{x}_0)$$
$$= \frac{q(\mathbf{x}_t|\mathbf{x}_{t-1})q(\mathbf{x}_{t-1}|\mathbf{x}_0)}{q(\mathbf{x}_t|\mathbf{x}_0)}$$
$$= \frac{\left(\beta_t \mathbf{x}_t + (1-\beta_t)\frac{\mathbf{1}}{K}\right) \odot \left(\alpha_{t-1}\mathbf{x}_0 + (1-\alpha_{t-1})\frac{\mathbf{1}}{K}\right)}{\mathbf{x}_t^\top \left(\alpha_t \mathbf{x}_0 + (1-\alpha_t)\frac{\mathbf{1}}{K}\right)}$$
$$= \frac{\alpha_t \mathbf{x}_t \odot \mathbf{x}_0 + \frac{1}{K}\beta_t(1-\alpha_{t-1})\mathbf{x}_t + \frac{1}{K}(1-\beta_t)\alpha_{t-1}\mathbf{x}_0 + \frac{1}{K^2}(1-\beta_t)(1-\alpha_{t-1})\mathbf{1}}{\alpha_t \mathbf{x}_t^\top \mathbf{x}_0 + \frac{1}{K}(1-\alpha_t)}. \tag{13}$$

Multinomial diffusion (Hoogeboom et al., 2021) learns a parameterized distribution $p_{\boldsymbol{\theta}}(\mathbf{x}_{t-1}|\mathbf{x}_t)$ to approximate $q(\mathbf{x}_{t-1}|\mathbf{x}_t, \mathbf{x}_0)$ at each time step, which is defined as $p_{\boldsymbol{\theta}}(\mathbf{x}_{t-1}|\mathbf{x}_t) = q(\mathbf{x}_{t-1}|\mathbf{x}_t, \tilde{\mathbf{x}}_0)$ with $\tilde{\mathbf{x}}_0 = f(\mathbf{x}_t; \boldsymbol{\theta})$ being the output by a Transformer model.

$$p_{\boldsymbol{\theta}}(\mathbf{x}_{t-1}|\mathbf{x}_t)$$
$$= \frac{\alpha_t \mathbf{x}_t \odot f(\mathbf{x}_t; \boldsymbol{\theta}) + \frac{1}{K}\beta_t(1-\alpha_{t-1})\mathbf{x}_t + \frac{1}{K}(1-\beta_t)\alpha_{t-1}f(\mathbf{x}_t; \boldsymbol{\theta}) + \frac{1}{K^2}(1-\beta_t)(1-\alpha_{t-1})\mathbf{1}}{\alpha_t \mathbf{x}_t^\top f(\mathbf{x}_t; \boldsymbol{\theta}) + \frac{1}{K}(1-\alpha_t)}. \tag{14}$$

## B  DERIVATION FOR PROPOSITION 3.1

In this section, we provide the derivation for Equation 2 based on Bayes' rule.

*Proof.* We denote $\mathbf{P}_t \in \mathbb{R}^{K \times K}$ as the probability transition matrix for the $t$-th step, where $[\mathbf{P}_t]_{ij} = p(\mathbf{x}_t = j|\mathbf{x}_{x-1} = i)$ and thus the probability distribution in the forward process can be described as $q(\mathbf{x}_t|\mathbf{x}_{t-1}) = \text{Categorical}\left(\mathbf{x}_t; \mathbf{P}_t^\top \mathbf{x}_{t-1}\right)$. It is then easy to see that

$$q(\mathbf{x}_t|\mathbf{x}_0) = \sum_{\mathbf{x}_{t-1}, \mathbf{x}_{t-2}, \ldots, \mathbf{x}_1} q(\mathbf{x}_t, \mathbf{x}_{t-1}, \mathbf{x}_{t-2}, \ldots, \mathbf{x}_1|\mathbf{x}_0) = \text{Categorical}\left(\mathbf{x}_t; \bar{\mathbf{P}}_t^\top \mathbf{x}_0\right)$$

with $\bar{\mathbf{P}}_t = \mathbf{P}_1 \mathbf{P}_2 \ldots \mathbf{P}_t$. Returning to our case where the forward transition takes the form $q(\mathbf{x}_t|\mathbf{x}_{t-1}) = \beta_t \mathbf{x}_{t-1} + (1 - \beta_t) q_{\text{noise}}$. The transition matrix can be represented by $\mathbf{P}_t =$

$\beta_t \mathbf{I} + (1 - \beta_t) \mathbf{1} q_{\text{noise}}^\top$, and thus $\bar{\mathbf{P}}_t = \mathbf{P}_1 \mathbf{P}_2 \dots \mathbf{P}_t = \alpha_t \mathbf{I} + (1 - \alpha_t) \mathbf{1} q_{\text{noise}}^\top$. Equipped with these results, we can proceed with the derivation as follows,

$$q(\mathbf{x}_{t-1} | \mathbf{x}_t, \mathbf{x}_0)$$

$$= \frac{q(\mathbf{x}_t | \mathbf{x}_{t-1}) q(\mathbf{x}_{t-1} | \mathbf{x}_0)}{q(\mathbf{x}_t | \mathbf{x}_0)} = \frac{\mathbf{P}_t \mathbf{x}_t \odot \bar{\mathbf{P}}_{t-1}^\top \mathbf{x}_0}{\mathbf{x}_t^\top \bar{\mathbf{P}}_t^\top \mathbf{x}_0}$$

$$= \frac{[\beta_t \mathbf{x}_t + (1 - \beta_t) \sigma_{\mathbf{x}_t} \mathbf{1}] \odot [\alpha_{t-1} \mathbf{x}_0 + (1 - \alpha_{t-1}) q_{\text{noise}}]}{\alpha_t \mathbf{x}_t^\top \mathbf{x}_0 + (1 - \alpha_t) \mathbf{x}_t^\top q_{\text{noise}}}$$

$$= \frac{\beta_t \alpha_{t-1} \mathbf{x}_t \odot \mathbf{x}_0 + \beta_t (1 - \alpha_{t-1}) \mathbf{x}_t \odot q_{\text{noise}} + (1 - \beta_t) \alpha_{t-1} \sigma_{\mathbf{x}_t} \mathbf{1} \odot \mathbf{x}_0 + (1 - \beta_t)(1 - \alpha_{t-1}) \sigma_{\mathbf{x}_t} \mathbf{1} \odot q_{\text{noise}}}{\alpha_t \mathbf{x}_t^\top \mathbf{x}_0 + (1 - \alpha_t) \mathbf{x}_t^\top q_{\text{noise}}}$$

$$= \frac{\beta_t \alpha_{t-1} \mathbf{x}_t \odot \mathbf{x}_0 + \beta_t (1 - \alpha_{t-1}) \sigma_{\mathbf{x}_t} \mathbf{x}_t + (1 - \beta_t) \alpha_{t-1} \sigma_{\mathbf{x}_t} \mathbf{x}_0 + (1 - \beta_t)(1 - \alpha_{t-1}) \sigma_{\mathbf{x}_t} q_{\text{noise}}}{\alpha_t \mathbf{x}_t^\top \mathbf{x}_0 + (1 - \alpha_t) \sigma_{\mathbf{x}_t}}.$$

Here we denote $\odot$ as element-wise product and $\sigma_{\mathbf{x}_t} := q_{\text{noise}}(\mathbf{u} = \mathbf{x}_t)$ to represent the probability of noise drawn from $q_{\text{noise}}$ being equal to $\mathbf{x}_t$. The need to differentiate between $\mathbf{x}_t$ and $\mathbf{x}_0$ emerges when we calculate $\mathbf{x}_t \odot \mathbf{x}_0$, which would be an all-zero vector $\mathbf{0}$ except that it would be one if $\mathbf{x}_t = \mathbf{x}_0$. Thus the computation of backward transition probabilities breaks down into two cases:

- If $\mathbf{x}_t = \mathbf{x}_0$, we have $\mathbf{x}_t \odot \mathbf{x}_0 = \mathbf{x}_t$, $\mathbf{x}_t \top \mathbf{x}_0 = 1$ and thus

$$q(\mathbf{x}_{t-1} | \mathbf{x}_t, \mathbf{x}_0)$$

$$= \frac{\beta_t \alpha_{t-1} \mathbf{x}_t + \beta_t (1 - \alpha_{t-1}) \sigma_{\mathbf{x}_t} \mathbf{x}_t + (1 - \beta_t) \alpha_{t-1} \sigma_{\mathbf{x}_t} \mathbf{x}_t + (1 - \beta_t)(1 - \alpha_{t-1}) \sigma_{\mathbf{x}_t} q_{\text{noise}}}{\alpha_t + (1 - \alpha_t) \sigma_{\mathbf{x}_t}}$$

$$= \frac{\beta_t \alpha_{t-1} + \beta_t (1 - \alpha_{t-1}) \sigma_{\mathbf{x}_t} + (1 - \beta_t) \alpha_{t-1} \sigma_{\mathbf{x}_t}}{\alpha_t + (1 - \alpha_t) \sigma_{\mathbf{x}_t}} \mathbf{x}_t + \frac{(1 - \beta_t)(1 - \alpha_{t-1}) \sigma_{\mathbf{x}_t}}{\alpha_t + (1 - \alpha_t) \sigma_{\mathbf{x}_t}} q_{\text{noise}}.$$

- If $\mathbf{x}_t \neq \mathbf{x}_0$, we have $\mathbf{x}_t \odot \mathbf{x}_0 = \mathbf{0}$, $\mathbf{x}_t \top \mathbf{x}_0 = 0$ and thus

$$q(\mathbf{x}_{t-1} | \mathbf{x}_t, \mathbf{x}_0) = \frac{\beta_t (1 - \alpha_{t-1}) \sigma_{\mathbf{x}_t} \mathbf{x}_t + (1 - \beta_t) \alpha_{t-1} \sigma_{\mathbf{x}_t} \mathbf{x}_0 + (1 - \beta_t)(1 - \alpha_{t-1}) \sigma_{\mathbf{x}_t} q_{\text{noise}}}{(1 - \alpha_t) \sigma_{\mathbf{x}_t}}$$

$$= \frac{\beta_t (1 - \alpha_{t-1}) \mathbf{x}_t + (1 - \beta_t) \alpha_{t-1} \mathbf{x}_0 + (1 - \beta_t)(1 - \alpha_{t-1}) q_{\text{noise}}}{1 - \alpha_t}$$

$$= \frac{(1 - \beta_t) \alpha_{t-1}}{1 - \alpha_t} \mathbf{x}_0 + \frac{\beta_t (1 - \alpha_{t-1})}{1 - \alpha_t} \mathbf{x}_t + \frac{(1 - \beta_t)(1 - \alpha_{t-1})}{1 - \alpha_t} q_{\text{noise}}$$

$$= \frac{(1 - \beta_t) \alpha_{t-1}}{1 - \alpha_t} \mathbf{x}_0 + \frac{1 - \alpha_{t-1}}{1 - \alpha_t} [\beta_t \mathbf{x}_t + (1 - \beta_t) q_{\text{noise}}]$$

$$= \frac{\alpha_{t-1} - \alpha_t}{1 - \alpha_t} \mathbf{x}_0 + \left(1 - \frac{\alpha_{t-1} - \alpha_t}{1 - \alpha_t}\right) [\beta_t \mathbf{x}_t + (1 - \beta_t) q_{\text{noise}}].$$

Putting them together, we arrive at the resulting formulation,

$$q(\mathbf{x}_{t-1} | \mathbf{x}_t, \mathbf{x}_0) = \begin{cases} \lambda_{t-1}^{(1)} \mathbf{x}_t + \left(1 - \lambda_{t-1}^{(1)}\right) q_{\text{noise}}, & \text{if } \mathbf{x}_t = \mathbf{x}_0 \\ \lambda_{t-1}^{(2)} \mathbf{x}_0 + \left(1 - \lambda_{t-1}^{(2)}\right) q_{\text{noise}}(\mathbf{x}_t), & \text{if } \mathbf{x}_t \neq \mathbf{x}_0. \end{cases}$$

Here $\lambda_{t-1}^{(1)} := 1 - \frac{(1 - \beta_t)(1 - \alpha_{t-1}) q_{\text{noise}}(\mathbf{u} = \mathbf{x}_t)}{\alpha_t + (1 - \alpha_t) q_{\text{noise}}(\mathbf{u} = \mathbf{x}_t)}$, $\lambda_{t-1}^{(2)} := \frac{\alpha_{t-1} - \alpha_t}{1 - \alpha_t}$, and $q_{\text{noise}}(\mathbf{x}_t) = \beta_t \mathbf{x}_t + (1 - \beta_t) q_{\text{noise}}$ denotes a noise distribution that interpolates between $\mathbf{x}_t$ and $q_{\text{noise}}$, both of which are possibly noisy.

**Generalization.** Similar to vanilla diffusion models, we can also derive backward transition processes with a gap $\Delta_t$; that is, we consider $q(\mathbf{x}_s | \mathbf{x}_t, \mathbf{x}_0)$ with $s = t - \Delta_t$. It can be easily seen that

$$q(\mathbf{x}_s | \mathbf{x}_t, \mathbf{x}_0) = \begin{cases} \lambda_s^{(1)} \mathbf{x}_t + \left(1 - \lambda_s^{(1)}\right) q_{\text{noise}}, & \text{if } \mathbf{x}_t = \mathbf{x}_0 \\ \lambda_s^{(2)} \mathbf{x}_0 + \left(1 - \lambda_s^{(2)}\right) q_{\text{noise}}(\mathbf{x}_t), & \text{if } \mathbf{x}_t \neq \mathbf{x}_0, \end{cases} \tag{15}$$

with $\lambda_s^{(1)} := 1 - \frac{(1 - \frac{\alpha_t}{\alpha_s})(1 - \alpha_s) q_{\text{noise}}(\mathbf{u} = \mathbf{x}_t)}{\alpha_t + (1 - \alpha_t) q_{\text{noise}}(\mathbf{u} = \mathbf{x}_t)}$ and $\lambda_s^{(2)} := \frac{\alpha_s - \alpha_t}{1 - \alpha_t}$. $\qquad \square$

## C  DERIVATIONS FOR THE ELBO OF RDMs

The following provides the derivation for the loss objective of RDMs. Specifically,

$$\log p(\mathbf{x}_0)$$

$$\geq \mathbb{E}_{q(\mathbf{x}_{1:T}, \mathbf{v}_{1:T}|\mathbf{x}_0)} \left[ \log \frac{p_{\boldsymbol{\theta}}(\mathbf{x}_0, \mathbf{x}_{1:T}, \mathbf{v}_{1:T})}{q(\mathbf{x}_{1:T}, \mathbf{v}_{1:T}|\mathbf{x}_0)} \right]$$

$$= \mathbb{E}_{q(\mathbf{x}_{1:T}, \mathbf{v}_{1:T}|\mathbf{x}_0)} \left[ \log \frac{p_{\boldsymbol{\theta}}(\mathbf{x}_0|\mathbf{x}_1) \prod_{t=2}^{T} p_{\boldsymbol{\theta}}(\mathbf{x}_{t-1}, \mathbf{v}_{t-1}|\mathbf{x}_t) p(\mathbf{x}_T, \mathbf{v}_T)}{\prod_{t=2}^{T} q(\mathbf{x}_{t-1}, \mathbf{v}_{t-1}|\mathbf{x}_t, \mathbf{x}_0) q(\mathbf{x}_T, \mathbf{v}_T|\mathbf{x}_0)} \right]$$

$$= \mathbb{E}_{q(\mathbf{x}_{1:T}, \mathbf{v}_{1:T}|\mathbf{x}_0)} \left[ \log p_{\boldsymbol{\theta}}(\mathbf{x}_0|\mathbf{x}_1) + \sum_{t=2}^{T} \log \frac{p_{\boldsymbol{\theta}}(\mathbf{x}_{t-1}, \mathbf{v}_{t-1}|\mathbf{x}_t)}{q(\mathbf{x}_{t-1}, \mathbf{v}_{t-1}|\mathbf{x}_t, \mathbf{x}_0)} + \log \frac{p(\mathbf{x}_T, \mathbf{v}_T)}{q(\mathbf{x}_T, \mathbf{v}_T|\mathbf{x}_0)} \right]$$

$$:= \mathcal{L}_1(\boldsymbol{\theta}) - \sum_{t=2}^{T} \mathcal{L}_t(\boldsymbol{\theta}) + \text{const.}$$

Here we denote $\mathcal{L}_1(\boldsymbol{\theta}) := \mathbb{E}_{q(\mathbf{x}_1|\mathbf{x}_0)} [\log p_{\boldsymbol{\theta}}(\mathbf{x}_0|\mathbf{x}_1)]$, and for $t > 1$, $\mathcal{L}_t(\boldsymbol{\theta}) := \mathbb{E}_{q(\mathbf{x}_t|\mathbf{x}_0)} [\text{KL}(q(\mathbf{x}_{t-1}, \mathbf{v}_{t-1}|\mathbf{x}_t, \mathbf{x}_0) \| p_{\boldsymbol{\theta}}(\mathbf{x}_{t-1}, \mathbf{v}_{t-1}|\mathbf{x}_t))]$. We can push the decomposition of $\mathcal{L}_t(\boldsymbol{\theta})$ for time step $t > 1$ further,

$$\mathcal{L}_t(\boldsymbol{\theta})$$

$$= \mathbb{E}_{q(\mathbf{x}_t|\mathbf{x}_0)} [\text{KL}(q(\mathbf{x}_{t-1}, \mathbf{v}_{t-1}|\mathbf{x}_t, \mathbf{x}_0) \| p_{\boldsymbol{\theta}}(\mathbf{x}_{t-1}, \mathbf{v}_{t-1}|\mathbf{x}_t))]$$

$$= \mathbb{E}_{q(\mathbf{x}_t|\mathbf{x}_0)} \left[ \sum_{\mathbf{x}_{t-1}, \mathbf{v}_{t-1}} q(\mathbf{x}_{t-1}, \mathbf{v}_{t-1}|\mathbf{x}_t, \mathbf{x}_0) \log \frac{q(\mathbf{x}_{t-1}, \mathbf{v}_{t-1}|\mathbf{x}_t, \mathbf{x}_0)}{p_{\boldsymbol{\theta}}(\mathbf{x}_{t-1}, \mathbf{v}_{t-1}|\mathbf{x}_t)} \right]$$

$$= \mathbb{E}_{q(\mathbf{x}_t|\mathbf{x}_0)} \left[ \sum_{\mathbf{x}_{t-1}, \mathbf{v}_{t-1}} q(\mathbf{x}_{t-1}|\mathbf{v}_{t-1}, \mathbf{x}_t, \mathbf{x}_0) q(\mathbf{v}_{t-1}) \left[ \log \frac{q(\mathbf{x}_{t-1}|\mathbf{v}_{t-1}, \mathbf{x}_t, \mathbf{x}_0)}{p_{\boldsymbol{\theta}}(\mathbf{x}_{t-1}|\mathbf{v}_{t-1}, \mathbf{x}_t)} + \log \frac{q(\mathbf{v}_{t-1})}{p_{\boldsymbol{\theta}}(\mathbf{v}_{t-1})} \right] \right]$$

$$= \mathbb{E}_{q(\mathbf{x}_t|\mathbf{x}_0)} \left[ \mathbb{E}_{q(\mathbf{v}_{t-1})} [\text{KL}(q(\mathbf{x}_{t-1}|\mathbf{v}_{t-1}, \mathbf{x}_t, \mathbf{x}_0) \| p_{\boldsymbol{\theta}}(\mathbf{x}_{t-1}|\mathbf{v}_{t-1}, \mathbf{x}_t))] + \text{KL}(q(\mathbf{v}_{t-1}) \| p_{\boldsymbol{\theta}}(\mathbf{v}_{t-1})) \right].$$

## D  DERIVATION FOR EQUATION 6 AND DISCUSSIONS

*Proof.* We first consider the loss objective at each step for *each token* at $n$-th position, which can be expanded as follows,

$$\mathcal{L}_t^n(\boldsymbol{\theta}) = \mathbb{E}_{q(\mathbf{x}_{t,n}|\mathbf{x}_{0,n})} \left[ \mathbb{E}_{q(\mathbf{v}_{t-1,n})} [\text{KL}(q(\mathbf{x}_{t-1,n}|\mathbf{v}_{t-1,n}, \mathbf{x}_{t,n}, \mathbf{x}_{0,n}) \| p_{\boldsymbol{\theta}}(\mathbf{x}_{t-1,n}|\mathbf{v}_{t-1,n}, \mathbf{x}_{t,n}))] \right].$$

Typically we draw a Monte Carlo sample $\mathbf{x}_{t,n} \sim q(\mathbf{x}_{t,n}|\mathbf{x}_{0,n})$ to estimate the outermost expectation above. For the inner term, recall that $b_{t,n} = \mathbf{1}_{\mathbf{x}_{t,n}=\mathbf{x}_{0,n}}$ and $q(\mathbf{x}_{t-1,n}|\mathbf{v}_{t-1,n}, \mathbf{x}_{t,n}, \mathbf{x}_{0,n})$ takes the form as

$$q(\mathbf{x}_{t-1,n}|\mathbf{v}_{t-1,n}, \mathbf{x}_{t,n}, \mathbf{x}_{0,n}) = \begin{cases} v_{t-1,n}^{(1)} \mathbf{x}_{t,n} + \left(1 - v_{t-1,n}^{(1)}\right) q_{\text{noise}}, & \text{if } b_{t,n} = 1 \\ v_{t-1,n}^{(2)} \mathbf{x}_{0,n} + \left(1 - v_{t-1,n}^{(2)}\right) q_{\text{noise}}(\mathbf{x}_{t,n}), & \text{if } b_{t,n} = 0. \end{cases}$$

Since we use the teacher-forcing approach that employs the same oracle $b_{t,n}$ for $p_{\boldsymbol{\theta}}(\mathbf{x}_{t-1,n}|\mathbf{v}_{t-1,n}, \mathbf{x}_{t,n})$ as well, it can also be written in a similar manner,

$$p_{\boldsymbol{\theta}}(\mathbf{x}_{t-1,n}|\mathbf{v}_{t-1,n}, \mathbf{x}_{t,n}) = \begin{cases} v_{t-1,n}^{(1)} \mathbf{x}_{t,n} + \left(1 - v_{t-1,n}^{(1)}\right) q_{\text{noise}}, & \text{if } b_{t,n} = 1 \\ v_{t-1,n}^{(2)} f(\mathbf{x}_{t,n}; \boldsymbol{\theta}) + \left(1 - v_{t-1,n}^{(2)}\right) q_{\text{noise}}(\mathbf{x}_{t,n}), & \text{if } b_{t,n} = 0. \end{cases}$$

The derivation then breaks down into two cases with respect to $b_{t,n}$:

**If $b_{t,n} = 1$.** In this case, $q(\mathbf{x}_{t-1,n}|\mathbf{v}_{t-1,n}, \mathbf{x}_{t,n}, \mathbf{x}_{0,n}) = p_{\boldsymbol{\theta}}(\mathbf{x}_{t-1,n}|\mathbf{v}_{t-1,n}, \mathbf{x}_{t,n}) = v_{t-1,n}^{(1)} \mathbf{x}_{t,n} + \left(1 - v_{t-1,n}^{(1)}\right) q_{\text{noise}}$. Since these two distributions become identical, this leads to zero KL divergence irrespective of $\mathbf{v}_{t-1,n}$ so that $\mathcal{L}_t(\boldsymbol{\theta}) = 0$;

**If** $b_{t,n} = 0$. In this scenario, we have $q(\mathbf{x}_{t-1,n}|b_{t-1}, \mathbf{x}_{t,n}, \mathbf{x}_{0,n}) = v_{t-1,n}^{(2)}\mathbf{x}_{0,n} + \left(1 - v_{t-1,n}^{(2)}\right) q_{\text{noise}}(\mathbf{x}_{t,n})$ and $v_{t-1,n}^{(2)} f(\mathbf{x}_{t,n}; \boldsymbol{\theta}) + \left(1 - v_{t-1,n}^{(2)}\right) q_{\text{noise}}(\mathbf{x}_{t,n})$. We then enumerate all the possible outcomes for $v_{t-1,n}^{(2)}$. If $v_{t-1,n}^{(2)} = 1$, $q(\mathbf{v}_{t-1,n}) = \lambda_{t-1}^{(2)}$ and

$$\text{KL}(q(\mathbf{x}_{t-1,n}|\mathbf{v}_{t-1,n}, \mathbf{x}_{t,n}, \mathbf{x}_{0,n}) \parallel p_{\boldsymbol{\theta}}(\mathbf{x}_{t-1,n}|\mathbf{v}_{t-1,n}, \mathbf{x}_{t,n})) = \text{KL}(\mathbf{x}_{0,n} \parallel f(\mathbf{x}_{t,n}; \boldsymbol{\theta}))$$
$$= -\mathbf{x}_{0,n}^{\top} \log f(\mathbf{x}_{t,n}; \boldsymbol{\theta}).$$

If $v_{t-1,n}^{(2)} = 0$, then $q(\mathbf{v}_{t-1,n}) = 1 - \lambda_{t-1}^{(2)}$ and

$$\text{KL}(q(\mathbf{x}_{t-1,n}|\mathbf{v}_{t-1,n}, \mathbf{x}_{t,n}, \mathbf{x}_{0,n}) \parallel p_{\boldsymbol{\theta}}(\mathbf{x}_{t-1,n}|\mathbf{v}_{t-1,n}, \mathbf{x}_{t,n})) = \text{KL}(q_{\text{noise}}(\mathbf{x}_{t,n}) \parallel q_{\text{noise}}(\mathbf{x}_{t,n}))$$
$$= 0.$$

Putting them together, we have

$$\mathbb{E}_{q(\mathbf{v}_{t-1,n})}\left[\text{KL}(q(\mathbf{x}_{t-1,n}|\mathbf{v}_{t-1,n}, \mathbf{x}_{t,n}, \mathbf{x}_{0,n}) \parallel p_{\boldsymbol{\theta}}(\mathbf{x}_{t-1,n}|\mathbf{v}_{t-1,n}, \mathbf{x}_{t,n}))\right]$$
$$= -\lambda_{t-1}^{(2)}\mathbf{x}_{0,n}^{\top} \log f(\mathbf{x}_{t,n}; \boldsymbol{\theta}) + (1 - \lambda_{t-1}^{(2)}) \cdot 0$$
$$= -\lambda_{t-1}^{(2)}\mathbf{x}_{0,n}^{\top} \log f(\mathbf{x}_{t,n}; \boldsymbol{\theta}).$$

Since each token is modeled conditionally independently, we can add all computed losses for each token, arriving at the final expression for the whole sequence,

$$\mathcal{L}_t(\boldsymbol{\theta}) = \sum_{n=1}^{N} \mathcal{L}_t^n(\boldsymbol{\theta}) = \mathbb{E}_{p_{\text{data}}(\mathbf{x}_{0,1:N}) \prod_{n=1}^{N} q(\mathbf{x}_{t,n}|\mathbf{x}_{0,n})}\left[-\lambda_{t-1}^{(2)}\sum_{n=1}^{N}(1 - b_{t,n})\mathbf{x}_{0,n}^{\top} \log f(\mathbf{x}_{t,n}; \boldsymbol{\theta})\right].$$

$\square$

**Connections to D3PMs.** The derived simplified loss objective bears some resemblance to that of absorbing diffusion in D3PMs (Austin et al., 2021), which also takes the form of a cross-entropy function over masked positions. However, our derivation arises from the developed reparameterization perspective and thus stems from a distinct motivation from D3PMs. In addition, our objective applies to a wide range of discrete diffusion processes, including those with multinomial noise, absorbing noise, or a mixture of both. This constitutes a non-trivial generalization of D3PMs, which only demonstrates that the cross-entropy representation is available for absorbing diffusion. Besides, our formulation explicitly elucidates the role of routing mechanisms in training, which provides insights into techniques that improve decoding quality.

# E ADDITIONAL IMPLEMENTATION DETAILS

This section describes the implementation details of our experiments.

## E.1 TASKS

**Machine Translation.** For experiments on machine translation, we consider three standard benchmarks:

- `IWSLT14 DE-EN` (Cettolo et al., 2014), which contains around 160K/7K/7K sentence pairs for training, validation, and testing, respectively. We build a joint vocabulary for the source and target language, resulting in 10152 Byte Pair Encoding (BPE; Sennrich et al., 2016) types.
- `WMT14 EN-DE` (Bojar et al., 2014) dataset consists of around 4.0M/3K/3K training/validation/testing pairs. The preprocessing follows Ghazvininejad et al. (2019) and yields a shared vocabulary with 40624 BPE types;
- `WMT16 EN-RO` (Bojar et al., 2016). We use the same data split from Lee et al. (2018) that comprises around 610K/2K/2K pairs. The vocabulary is shared between the source and target sides with 34976 joint BPE types.

Table 5: Comparisons among different evaluation metrics on `WMT14 EN-DE`.

| Model | Iterations | Tokenized BLEU | sacreBLEU | COMET |
|---|---|---|---|---|
| Auto-regressive | n.a. | 27.53 | 26.5 | 0.8238 |
| RDM-multinomial | 10 | 25.63 | 24.1 | 0.7808 |
| | 16 | 25.64 | 24.2 | 0.7937 |
| RDM-absorbing | 10 | 26.96 | 25.2 | 0.8082 |
| | 16 | 27.58 | 26.2 | 0.8288 |

We operate on original data for all translation tasks and do *not* adopt knowledge distillation (Kim & Rush, 2016; Gu et al., 2018) that replaces the target side of training data with outputs generated by a pre-trained autoregressive Transformer.

For evaluation, we report tokenized BLEU (Papineni et al., 2002) scores applied with compound split post-processing to facilitate comparison. In addition, we also compute sacreBLEU (Papineni et al., 2002; Post, 2018) (signature: `nrefs:1|case:mixed|eff:no|tok:13a|smooth:exp|version:2.0.0`) and COMET (Rei et al., 2020) (with version `2.0.0` and model `Unbabel/wmt22-comet-da`) scores on `WMT14 EN-DE` test set, as presented in Table 5. It can be seen that both sacreBLEU and COMET reveal a trend similar to that of tokenized BLEU scores.

**Question Generation and Paraphrasing.** For both `QG` and `QQP` tasks, we use the same data split pre-processed as in Gong et al. (2022):

- Question Generation (`QG`) with the Quasar-T dataset (Dhingra et al., 2017), containing around 117K/2K/10K pairs for training/validation/testing, respectively.
- Paraphrasing with Quora Question Pairs (`QQP`). This dataset comprises around 145K/2K/2.5K training/validation/testing question pairs.

Following Gong et al. (2022), we use WordPiece tokenization as in BERT (Devlin et al., 2019) and obtain a vocabulary of size 30522 for both tasks.

### E.2 ARCHITECTURES

- We employ the Transformer-base architecture (Vaswani et al., 2017) for WMT experiments, while for `IWSLT14 DE-EN`,`QG`, and `QQP` tasks we use a smaller Transformer model. Note that all self-attention blocks with the model are bi-directional and do not use causal masks.
- We adopt a length prediction module (Ghazvininejad et al., 2019) on top of the Transformer encoder to propose target length candidates for the generated sequence. Given the source input, we first run the Transformer encoder to obtain the encoder's hidden representation, which is averaged and passed to a linear layer to output the length scores.
- The timestep embedding is obtained by first projecting the input timestep $t$ with sinusoidal encodings and then passing it through a two-layer MLP.
- We adopt *concatenated* instead of additive position encodings, which is shown to enhance the positional information and produce better performance in the context of machine translation (Huang et al., 2022).

The detailed configuration for the neural network is listed in Table 6.

### E.3 TRAINING

- We allocate a large number of diffusion time steps for training, such as 50 or 100. We found the number of diffusion steps *in training* does not affect the performance too much. Note that decoding can be performed with an arbitrary number of iterations by choosing the appropriate step size $\Delta t > 1$. That is, we can decode by sampling $\mathbf{x}_{t-\Delta t} \sim p_{\boldsymbol{\theta}}(\mathbf{x}_{t-\Delta t}|\mathbf{x}_t)$, following a similar treatment as Equation 15.
- Modern Transformer models usually process input sequences that are associated with some special symbols, such as the begin-of-sentence symbol <bos>, eos-of-sentence symbol <eos>, padding

Table 6: The hyper-parameter configuration for machine translation.

| Hyper-parameter | WMT14 EN-DE | WMT16 EN-RO | IWSLT14 DE-EN | QG | QQP |
|---|---|---|---|---|---|
| Number of transformer encoder layers | 6 | 6 | 6 | 6 | 6 |
| Number of transformer decoder layers | 6 | 6 | 6 | 6 | 6 |
| Hidden size | 512 | 512 | 512 | 512 | 512 |
| hidden size in FFN | 2048 | 2048 | 1024 | 1024 | 1024 |
| Number of attention heads | 8 | 8 | 4 | 8 | 8 |
| Maximum number of tokens in a batch | 128K | 32K | 4K | – | – |
| Maximum number of sentences in a batch | – | – | – | 256 | 256 |
| Number of training steps | 300K | 120K | 300K | 70K | 70K |
| Number of warm-up steps | 10K | 15K | 30K | 10K | 10K |
| Weight decay rate | 0.01 | 0.01 | 0.01 | 0.01 | 0.01 |
| Peak Learning Rate | 0.0005 | 0.0005 | 0.0005 | 0.0005 | 0.0005 |
| Label Smoothing | 0.1 | 0.1 | 0.1 | 0.1 | 0.1 |
| Learning rate decay | Inverse square root | Inverse square root | Inverse square root | Inverse square root | Inverse square root |
| Optimizer | Adam | Adam | Adam | Adam | Adam |
| Dropout | 0.1 | 0.3 | 0.3 | 0.2 | 0.2 |
| Gradient Clipping Norm | – | – | – | 1.0 | 1.0 |

Table 7: BLEU scores on IWSLT14 DE-EN test set with/without conditioned training. The results are evaluated under different diffusion models with 10 decoding iterations. Default decoding strategies are adopted for these models: vanilla multinomial or absorbing diffusion uses vanilla decoding, while reparameterized discrete diffusion models adopt improved decoding.

| Conditioned training | Absorbing | Multinomial | RDM-absorbing | RDM-multinomial |
|---|---|---|---|---|
| ✗ | 28.32 | 21.28 | 33.73 | 31.99 |
| ✓ | **29.67** | **23.58** | **33.91** | **32.23** |

symbol <pad>, and so on. We found it beneficial to treat these special symbols as normal tokens in vanilla/reparameterized absorbing diffusion (and thus these symbols can be noised), but this treatment leads to much worse performance in vanilla/reparameterized multinomial diffusion.

- We also adopt conditioned training (details in Appendix F) to further improve the model, which leads to consistent improvements upon vanilla training. Its effect is ablated in Table 7. In particular, conditioned training leads to almost 2 BLEU improvements over vanilla diffusion processes but the gain becomes marginal for our reparameterized variants.

The detailed configuration for the optimization hyper-parameters is listed in Table 6.

### E.4 DECODING

- Note that we train a neural network $f(\cdot; \boldsymbol{\theta})$ to approximate $\mathbf{x}_0$, which is a softmax-normalized probability vector. There are several ways to decode a token from the probability vector, such as simply taking its argmax position or performing sampling with temperatures $\tau$. Empirically, we find a low temperature $\tau = 0.1$, or simply the argmax works well across tasks, and use the argmax approach by default; Nevertheless, the diversity of generated sentences can be improved by adopting a larger temperature as well.

- Like common diffusion models, we use an exponential moving average (EMA) to track the model parameters with a decay rate of 0.9999. We also average model parameters among the five last checkpoints for generation, following standard practices in machine translation (Vaswani et al., 2017).

- For translation experiments, we use 5 length candidates, decode them in parallel, and select the sequence with the highest model score as the final output. For question generation and paraphrasing, we follow DiffuSeq (Gong et al., 2022) to use MBR decoding with 10 samples to ensure a head-to-head comparison. The candidates in MBR decoding are generated in the following manner: first selecting 3 length candidates, and then sampling 3,3, and 4 additional sentences for each length size, resulting in 10 candidates in total.

- We also investigate several components of the adaptive decoding algorithm (§4.3) and provide more implementation details below:

  - The top-$k$ selection mechanism in Equation 7, which is deterministic by design, can also be made stochastic. In particular, instead of directly selecting those tokens with top-$k$ largest scores, we first add Gumbel noise to the score $s_{t,n}$ of each token, and then fetch the top-$k$ tokens with

the largest perturbed scores. This is inspired by previous work that aims to sample multiple items from a set without replacement (Vieira, 2014; Kool et al., 2019; 2020); however, this simple approach brings several benefits in that the selection of $k$ tokens from the sequence could involve extra randomness, and this might be helpful for exploration during decoding.

- Recall that during decoding, the role of $\mathbf{v}_{t-1,n}^{(1)}$ is to indicate whether the token can remain in the denoised state, while $\mathbf{v}_{t-1,n}^{(2)}$ is used to denoise the token that is currently noisy. In Equation 7, both of them would be set to 1 as long as the $n$-th token belongs to the top-$k$ set. However, we observe that $\mathbf{v}_{t-1,n}^{(1)}$ and $\mathbf{v}_{t-1,n}^{(2)}$ can also be generated differently. For example, one might adopt a more conservative approach, where already denoised tokens rarely or never turn back to the noise. We implemented a strategy to achieve this by imposing more constraints over $\mathbf{v}_{t-1,n}^{(1)}$: $\mathbf{v}_{t-1,n}^{(i)} = \mathbf{1}_{(n \in \mathcal{P}_{t-1}) \vee (s_{t,n} > s_{t+1,n}) \vee (\mathbf{x}_{t,n} \neq \mathbf{x}_{t+1,n})}$, where we set $\mathbf{v}_{t-1,n}^{(1)} = 0$ only when its corresponding token score is not in the top-$k$ set and indeed becomes smaller than the previous iteration. Intuitively, this means the denoised tokens should remain as denoised most time, except that the Transformer model becomes less confident and requires re-prediction. This is one of many possible approaches to achieving such control, as our framework allows us to do such conditioning flexibly; we find this strategy works sometimes better than the vanilla approach, especially on IWSLT14 DE-EN dataset.
- Another important hyper-parameter during decoding is $k$, the number of tokens to be in denoised states at each iteration for our discriminative routing mechanism (§4.3). To ensure that the degree of noise decreases as the generation process proceeds, we schedule $k$ to increase from 1 to $N$ monotonically as the diffusion step $t$ goes from $T$ to 1. We set $k$ to follow either a $\{\text{cosine}, \text{linear}\}$ scheme based on the development set performance. The cosine strategy yields $k = \lfloor \cos \frac{\pi t}{2T} \cdot N \rfloor$, while the linear variant gives $k = \lfloor \left(1 - \frac{t}{T}\right) \cdot N \rfloor$.

Based on our preliminary experiments, we discerned that while these components indeed have an influence on task performance, their impact is relatively minor compared to the primary improvements (e.g., reweighted training and adaptive decoding), as elucidated in the main paper.

- To perform conditional generation, we delegate the full task of conditioning to the encoder-decoder architecture. That is, instead of designing complicated guidance to condition the diffusion probabilistic process, we treat the conditioning information as the input of the Transformer encoder. This simple strategy is found to work well in practice.

## F  EXTENSION: IMPROVED TRAINING WITH CONDITIONING

Training discrete diffusion models usually involves a heavy amount of randomness. For instance, at each training iteration, one has to sample a time step and corrupt a random subset of sequence tokens for denoising. To control the introduced variance, we adopt a simple yet effective conditioning strategy that uses multiple samples to perform training. The key idea is conceptually simple: we start with sampling two *i.i.d.* time steps $s, t \sim \text{Uniform}(T)$ (without the loss of generality, we assume $s < t$). After that, we draw $\mathbf{x}_t \sim q(\mathbf{x}_t|\mathbf{x}_0)$ as usual, but condition the sample at step $s$ by drawing from $q(\mathbf{x}_s|\mathbf{x}_0) = \mathbb{E}_{q(\mathbf{x}_t|\mathbf{x}_0)}[q(\mathbf{x}_s|\mathbf{x}_t, \mathbf{x}_0)] \approx q(\mathbf{x}_s|\mathbf{x}_t, \mathbf{x}_0)$. The losses (Equation 6) at step $s$ and $t$ are then estimated individually and averaged to obtain the final training objective. In case $s = t$, we simply drop the conditioning and sample $\mathbf{x}_s \sim (\mathbf{x}_s|\mathbf{x}_0)$ instead.

This method utilizes multiple samples to estimate the loss objective while remaining unbiased. To see this, note that

$$- (\mathcal{L}_s + \mathcal{L}_t)$$
$$= \mathbb{E}_q \left[ \log \frac{q(\mathbf{x}_{s-1}|\mathbf{x}_s, \mathbf{x}_0)}{p_{\boldsymbol{\theta}}(\mathbf{x}_{s-1}|\mathbf{x}_s)} + \log \frac{q(\mathbf{x}_{t-1}|\mathbf{x}_t, \mathbf{x}_0)}{p_{\boldsymbol{\theta}}(\mathbf{x}_{t-1}|\mathbf{x}_t)} \right]$$
$$= \mathbb{E}_{q(\mathbf{x}_{s-1},\mathbf{x}_s,\mathbf{x}_{t-1},\mathbf{x}_t|\mathbf{x}_0)} \left[ \log \frac{q(\mathbf{x}_{s-1}|\mathbf{x}_s, \mathbf{x}_0)}{p_{\boldsymbol{\theta}}(\mathbf{x}_{s-1}|\mathbf{x}_s)} + \log \frac{q(\mathbf{x}_{t-1}|\mathbf{x}_t, \mathbf{x}_0)}{p_{\boldsymbol{\theta}}(\mathbf{x}_{t-1}|\mathbf{x}_t)} \right]$$
$$= \mathbb{E}_{q(\mathbf{x}_{s-1},\mathbf{x}_s,\mathbf{x}_{t-1},\mathbf{x}_t|\mathbf{x}_0)} \left[ \log \frac{q(\mathbf{x}_{s-1}|\mathbf{x}_s, \mathbf{x}_0)}{p_{\boldsymbol{\theta}}(\mathbf{x}_{s-1}|\mathbf{x}_s)} \right] + \mathbb{E}_{q(\mathbf{x}_{t-1},\mathbf{x}_t|\mathbf{x}_0)} \left[ \log \frac{q(\mathbf{x}_{t-1}|\mathbf{x}_t, \mathbf{x}_0)}{p_{\boldsymbol{\theta}}(\mathbf{x}_{t-1}|\mathbf{x}_t)} \right]$$
$$= \mathbb{E}_{q(\mathbf{x}_t|\mathbf{x}_0)q(\mathbf{x}_s|\mathbf{x}_t, \mathbf{x}_0)} \left[ \text{KL}(q(\mathbf{x}_{s-1}|\mathbf{x}_s, \mathbf{x}_0) \| p_{\boldsymbol{\theta}}(\mathbf{x}_{s-1}|\mathbf{x}_s)) \right] +$$
$$\mathbb{E}_{q(\mathbf{x}_t|\mathbf{x}_0)} \left[ \text{KL}(q(\mathbf{x}_{t-1}|\mathbf{x}_t, \mathbf{x}_0) \| p_{\boldsymbol{\theta}}(\mathbf{x}_{t-1}|\mathbf{x}_t)) \right].$$

---

**Algorithm 3** Training RDMs with Conditioning

---

**Input:** neural network $f(\cdot; \boldsymbol{\theta})$, data distribution $p_{\text{data}}(\mathbf{x}_{0,1:N})$, and a specified reweighting scalar $\lambda_{t-1}, \lambda_{s-1}$.
**Output:** model parameters $\boldsymbol{\theta}$.
**repeat**
    Draw $\mathbf{x}_{0,1:N} \sim p_{\text{data}}(\mathbf{x}_{0,1:N})$;
    Draw $s \in \text{Uniform}(\{1, \ldots, T\})$;
    Draw $t \in \text{Uniform}(\{1, \ldots, T\})$;
    Swap $t$ and $s$ if necessary so that $s \leq t$;
    **for** $n = 1, 2, \ldots, N$ **do**
        Draw $\mathbf{x}_{t,n} \sim q(\mathbf{x}_{t,n}|\mathbf{x}_{0,n})$;
        Let $b_{t,n} = \mathbf{1}_{\mathbf{x}_{t,n}=\mathbf{x}_{0,n}}$;
        **if** $s = t$ **then**
            Draw $\mathbf{x}_{s,n} \sim q(\mathbf{x}_{s,n}|\mathbf{x}_{0,n})$;
        **else**
            Draw $\mathbf{x}_{s,n} \sim q(\mathbf{x}_{s,n}|\mathbf{x}_{t,n}, \mathbf{x}_{0,n})$;
        **end if**
        Let $b_{s,n} = \mathbf{1}_{\mathbf{x}_{s,n}=\mathbf{x}_{0,n}}$;
    **end for**
    $\mathcal{L}_t(\boldsymbol{\theta}) = -\lambda_{t-1}\sum_{n=1}^{N}(1-b_{t,n})\mathbf{x}_{0,n}^{\top}\log f(\mathbf{x}_{t,n}; \boldsymbol{\theta})$;
    $\mathcal{L}_s(\boldsymbol{\theta}) = -\lambda_{s-1}\sum_{n=1}^{N}(1-b_{s,n})\mathbf{x}_{0,n}^{\top}\log f(\mathbf{x}_{s,n}; \boldsymbol{\theta})$;
    Compute $\mathcal{L}(\boldsymbol{\theta}) = \frac{1}{2}(\mathcal{L}_s(\boldsymbol{\theta}) + \mathcal{L}_t(\boldsymbol{\theta}))$;
    Minimize $\mathcal{L}(\boldsymbol{\theta})$ with respect to $\boldsymbol{\theta}$;
**until** converged

---

This conditioned training brings several benefits. On the one hand, it introduces explicit coupling between $\mathbf{x}_s$ and $\mathbf{x}_t$, which can be seen as an application of Rao-blackwellization and constrains the degree of randomness while still maintaining unbiasedness; on the other hand, as we deal with samples from the simulated backward transition $q(\mathbf{x}_s|\mathbf{x}_t, \mathbf{x}_0)$ instead of $q(\mathbf{x}_s|\mathbf{x}_0)$, this formulation aligns better with the generation process. In practice, this technique can be applied to most existing diffusion processes, only amounts to double the batch size within a single model forward pass, and consistently brings large empirical improvements over vanilla discrete diffusion baselines (Table 7). However, the gains become marginal when switched to our reparameterized variants. We hypothesize that this is due to insufficient training in vanilla discrete diffusion models, which is already alleviated in our improved training scheme.

## G    ADDITIONAL EXPERIMENTAL RESULTS

### G.1    ADDITIONAL EXPERIMENTS FOR QUESTION AND PARAPHRASE GENERATION

Table 8 presents the comparison between text diffusion models under different number of candidate samples. We notice that DiffuSeq benefits slightly more from large sample sets (e.g., when the sample size $m$ increases from 1 to 10) than RDMs. We attribute this to the possibility that adding Gaussian noise to token embeddings in DiffuSeq might lead to more diverse samples. This helps make better use of the MBR decoding, indicating that there might be room to improve RDMs to leverage multiple decodes. Nevertheless, RDMs still achieve better performance than DiffuSeq across both cases of single and multiple samples. Note that due to the limited computation resources, our reproduction of DiffuSeq (Gong et al., 2022) adopts a smaller Transformer model with around 36M parameters (the same as our setting) and runs with a smaller batch size of 256, thus resulting in slightly worse results than those reported.

### G.2    ADDITIONAL EXPERIMENTS FOR RUNTIME COMPARISON

Table 9 compares the decoding runtime between RDMs and prior non-autoregressive baselines, namely CMLM (Ghazvininejad et al., 2019). Note that both CMLM and RDMs are implemented with the same codebase `fairseq` (Ott et al., 2019), and the statistics are calculated as the wall time

Table 8: Comparisons among different text generators on QG and QQP tasks. Numbers are taken from Gong et al. (2022). † denotes results due to our implementation. $m$ denotes the number of samples used for MBR decoding. RDM variants are run with 10 iterations.

| Task | Model | BLEU ↑ | ROUGE-L ↑ | BERTScore ↑ | Dist-1↑ |
|---|---|---|---|---|---|
| QG | Transformer-base | 0.1663 | 0.3441 | 0.6307 | 0.9309 |
| | GPT2-base FT | 0.0741 | 0.2714 | 0.6052 | 0.9602 |
| | GPT2-large FT | 0.1110 | 0.3215 | 0.6346 | **0.9670** |
| | GPVAE-T5 | 0.1251 | 0.3390 | 0.6308 | 0.9381 |
| | NAR-LevT | 0.0930 | 0.2893 | 0.5491 | 0.8914 |
| | DiffuSeq | 0.1731 | **0.3665** | 0.6123 | 0.9056 |
| | DiffuSeq† ($m=1$) | 0.1405 | 0.3343 | 0.5783 | 0.9109 |
| | RDM-absorbing† ($m=1$) | 0.1699 | 0.3517 | 0.6286 | 0.9098 |
| | RDM-multinomial† ($m=1$) | 0.1768 | 0.3559 | 0.6305 | 0.9081 |
| | DiffuSeq† ($m=10$) | 0.1569 | 0.3561 | 0.5945 | 0.9062 |
| | RDM-absorbing† ($m=10$) | 0.1791 | 0.3565 | **0.6393** | 0.9202 |
| | RDM-multinomial† ($m=10$) | **0.1802** | 0.3550 | 0.6310 | 0.9082 |
| QQP | Transformer-base | **0.2722** | 0.5748 | 0.8381 | 0.9748 |
| | GPT2-base FT | 0.1980 | 0.5212 | 0.8246 | 0.9798 |
| | GPT2-large FT | 0.2059 | 0.5415 | 0.8363 | 0.9819 |
| | GPVAE-T5 | 0.2409 | 0.5886 | 0.8466 | 0.9688 |
| | NAR-LevT | 0.2268 | 0.5795 | 0.8344 | 0.9790 |
| | DiffuSeq | 0.2413 | 0.5880 | 0.8365 | 0.9807 |
| | DiffuSeq† ($m=1$) | 0.1845 | 0.5284 | 0.7936 | 0.9739 |
| | RDM-absorbing† ($m=1$) | 0.2336 | 0.5789 | 0.8374 | 0.9805 |
| | RDM-multinomial† ($m=1$) | 0.2291 | 0.5725 | 0.8366 | 0.9802 |
| | DiffuSeq† ($m=10$) | 0.2371 | 0.5835 | 0.8327 | 0.9818 |
| | RDM-absorbing† ($m=10$) | 0.2510 | **0.5945** | **0.8472** | **0.9849** |
| | RDM-multinomial† ($m=10$) | 0.2498 | 0.5886 | 0.8466 | 0.9817 |

Table 9: Decoding runtime comparison between the non-autoregressive baseline CMLM and RDMs.

| Iteration | CMLM | | RDM-absorbing | | RDM-multinomial | |
|---|---|---|---|---|---|---|
| | BLEU | Runtime | BLEU | Runtime | BLEU | Runtime |
| 2 | 19.73 | 13.4s | 21.00 | 12.7s | 21.43 | 14.4s |
| 4 | 22.91 | 17.1s | 24.26 | 18.2s | 24.05 | 18.1s |
| 10 | 24.89 | 32.2s | 26.96 | 31.5s | 25.63 | 34.7s |
| 16 | 25.00 | 44.7s | 27.58 | 44.9s | 25.64 | 46.8s |

to decode the entire WMT14 EN-DE test set on one NVIDIA GeForce RTX 3090 GPU with 50 batch size and 16 iteration steps, averaged by 10 runs. Under this setting, we observe that the primary factor affecting decoding runtime among non-autoregressive baselines and our diffusion models is the number of Transformer decoder calls.

### G.3 ADDITIONAL EXPERIMENTS FOR OPEN-ENDED TEXT GENERATION

In this section, we further explore the generative capabilities of RDMs in open-ended text generation. In particular, we conduct experiments on the Wikitext-103 dataset (Merity et al., 2016) from the Wikipedia domain, and recruit the following metrics for automatic evaluation according to prior research on open-ended text generation (Li et al., 2022a; Su & Collier, 2022): **1)** Diversity, which measures the generation repetition at different $n$-gram levels; **2)** MAUVE (Pillutla et al., 2021), evaluating the distance of token distributions between the generated and human-written text on the test set; and **3)** Coherence, calculating the average-token log likelihood under a well-trained language model, which is set to OPT-2.7B (Zhang et al., 2022). We refer readers to Su & Collier (2022) for more technical details of these metrics.

We train three different models for comparison: auto-regressive language models, vanilla discrete diffusion models, and our RDMs. All of these models have approximately 430M parameters and are trained with $100k$ steps on the Wikitext-103 training set. Both auto-regressive language models and discrete diffusion models here adopt the same decoder-only Transformers following the Llama architecture (Touvron et al., 2023), except that discrete diffusion models remove the use of causal masks in self-attention blocks and introduce an additional lightweight time-step embedding for proper conditioning. During training, the maximum sequence length and the batch size are set to 256 and 128, respectively, where shorter sequences are packed together. The Adam (Kingma & Ba,

Table 10: Automatic evaluation results on Wikitext-103 with different text generation models.

| Model | Iteration | Diversity(%) | MAUVE(%) | Coherence |
|---|---|---|---|---|
| Autoregressive LM | n.a. | 94.07 | **95.71** | **-4.54** |
| Absorbing Diffusion | 16 | **98.04** | 47.75 | -7.04 |
| | 64 | 96.93 | 64.61 | -6.59 |
| | 100 | 97.01 | 75.70 | -6.38 |
| RDM-absorbing | 16 | 97.32 | 62.32 | -6.04 |
| | 64 | 96.84 | 79.14 | -5.68 |
| | 100 | 96.80 | 91.66 | -5.15 |

Table 11: Generation examples of open-ended text generation on Wikitext-103 with different text models. The underlined text denotes the prefix prompt for generation.

| Model | Decodes |
|---|---|
| Autoregressive LM | **The route of what became US 2 was used as part of two Indian trails before European settlers came to the UP, and as part of the Michigan segments** of the Great Northern Railway and Chicago Railway . The various segments of the UP were later often referred to other western Ohio routes such as the Mid @-@ Continental Route , I @-@ 280 and the Western Interstate route , which went through eastern Ohio into eastern Ohio near Knolls Creek . |
| Absorbing Diffusion - 64 Steps | **The route of what became US 2 was used as part of two Indian trails before European settlers came to the UP, and as part of the Michigan segments** . When the current Detroit segment was completed , both in 1956 and 1973 and removed from the state . |
| RDM-absorbing - 64 Steps | **The route of what became US 2 was used as part of two Indian trails before European settlers came to the UP, and as part of the Michigan segments** of this section , it is included in the 1930 State Highway construction portion that exceeds only $generally at$ 4 @.@ 5 million in 1935 . In January 1936 , another state highway was created . The B̈ig Pine Highway f̈lew by DVP was so named State Highway 7 . |

2014) optimizer is used, and the learning rate is set to 3e-4 with the cosine scheduler. To facilitate evaluation on *open-ended* generation, we follow previous practices (Li et al., 2022a) and condition the generation of different models on test prefix prompts with a fixed length of 32. We limit the maximum generation length to 256 and truncate the generated output for each test case to the first 128 tokens for subsequent evaluation. For diffusion models, we the initial sequence length to 256 and truncate all content after the first <eos> token upon the iterative process finishes. For all models, we generate samples at the temperature of 1.0 by nucleus sampling (Holtzman et al., 2020) with top-$p$ 0.95. Table 10 demonstrates the comparison among autoregressive language models, vanilla discrete diffusion models, and our RDMs in the task of open-ended text generation. We observe that while discrete diffusion models generally lag behind auto-regressive LMs, RDMs effectively reduce the gap, scale well with the number of iterations, and achieve competitive performance with auto-regressive LMs while exhibiting greater generation variety. Generation examples can be found in Table 11.

## G.4 ADDITIONAL ABLATION STUDY FOR TRANSLATION TASKS

This section presents additional plots (Figure 3) that visualize the effect of different components in our model.

## G.5 EXTENDED QUALITATIVE ANALYSIS

This section provides a more comprehensive qualitative analysis of different diffusion models, including several generated samples as in Tables 12 to 14.

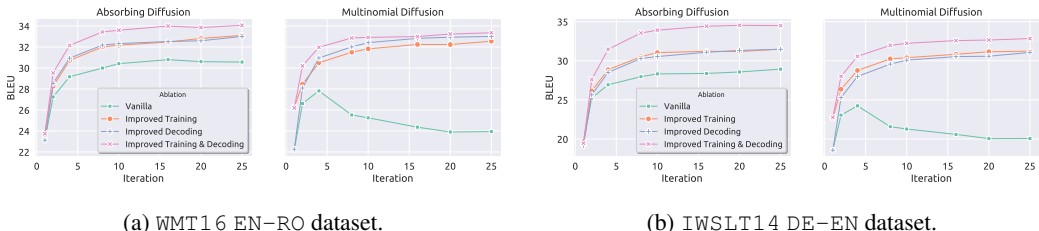

(a) `WMT16 EN-RO` dataset.  (b) `IWSLT14 DE-EN` dataset.

Figure 3: The ablation study of improved training and decoding strategies for both absorbing diffusion and multinomial diffusion on `WMT16 EN-RO` and `IWSLT14 DE-EN` test sets.

Table 12: A snapshot of all iterations for qualitative samples of test paraphrases generated from different diffusion models on `QQP` dataset. $^\ddagger$ texts are truncated to fit into the table. Words are in lower case. <M> stands for mask states, and ## denotes the sub-word tokenization artifacts.

| | # Iter. | Decodes |
|---|---|---|
| **Absorbing** | 0 | ∘ <M> <M> <M> <M> <M> <M> <M> <M> <M> <M> <M> <M> <M> <M> <M> |
| | 1 | ∘ <M> <M> <M> <M> <M> ca <M> ca <M> <M> ca <M> <M> <M> <M> |
| | 2 | ∘ <M> <M> <M> <M> <M> ca <M> ca <M> <M> ca <M> <M> <M> <M> |
| | 3 | ∘ <M> <M> <M> <M> <M> ca - ca cp <M> ca <M> <M> exam <M> |
| | 4 | ∘ <M> can <M> prepare <M> ca - ca cp <M> ca <M> ##t exam ? |
| | 5 | ∘ how can i prepare for ca - ca cp ##t ca cp ##t exam ? |
| **RDM-absorbing** | 0 | ∘ <M> <M> <M> <M> <M> <M> <M> <M> <M> <M> <M> <M> <M> <M> <M> |
| | 1 | ∘ how <M> i <M> <M> <M> <M> <M> <M> <M> <M> <M> <M> <M> ? |
| | 2 | ∘ how <M> i prepare for ca <M> ##t <M> <M> <M> <M> <M> ? |
| | 3 | ∘ how <M> i prepare for ca cp ##t <M> <M> <M> months months ? |
| | 4 | ∘ how <M> i prepare for ca cp ##t exam in two months <M> ? |
| | 5 | ∘ how can i prepare for ca cp ##t exam in two months left ? |
| **Multinomial** | 0 | ∘ glossy [unused448] raymond manga subjective questioning suriname masonic listen explored |
| | 1 | ∘ how can i prepare for ca cp ##t months ? |
| | 2 | ∘ how can i prepare for ca cp ##t months ? |
| | 3 | ∘ how can i prepare for ca cp ##t months ? |
| | 4 | ∘ how can i prepare for ca cp ##t months ? |
| | 5 | ∘ how can i prepare for ca cp ##t months ? |
| **RDM-multinomial** | 0 | ∘ consonants ##nin leading elegance 406 173 militant teams ##nin dyke thee seafood |
| | 1 | ∘ how residues i ##fighting sentences malaysian jenkins remembers transatlantic universite ##rp monarch ? |
| | 2 | ∘ how can i cyril for malaysian jenkins goldberg transatlantic in relationships pursuing ? |
| | 3 | ∘ how can i chu for ca fashionable ##t exam in clerks months ? |
| | 4 | ∘ how can i prepare for ca cp ##t exam in 2 months ? |
| | 5 | ∘ how can i prepare for ca cp ##t exam in 2 months ? |
| **DiffuSeq** | 0 | ∘ defective thereby evaluation michaels fragments primal electrically aground hostilities$^\ddagger$ |
| | 10 | ∘ simulcast candidacy ##bner [unused106] ##wide subgenus dangerously sincerity resolving migrated menon ##lase $^\ddagger$ |
| | 100 | ∘ westphalia ##tracted universite ##erly reissued neglect showcased [unused574] slade$^\ddagger$ |
| | 250 | ∘ souza electronically compliant gerard priority townships ##neo hidalgo [unused574] $^\ddagger$ |
| | 500 | ∘ spikes peptide ##ales borneo makeshift moi rebelled neglect textual 1899 erasmus publishes $^\ddagger$ |
| | 750 | ∘ i reduces griffin ##ales bukit makeshift moi ##sław mcbride how ministries $^\ddagger$ |
| | 1000 | ∘ i [unused582] to gazing monterrey makeshift ca wastewater norton , how ministries can$^\ddagger$ |
| | 1001 | ∘ i [unused582] to gazing monterrey makeshift ca iata norton , how ministries can $^\ddagger$ |
| | 1002 | ∘ i [unused582] to gazing monterrey makeshift ca iata norton , how ministries can $^\ddagger$ |
| | 1003 | ∘ i [unused582] to gazing monterrey makeshift ca iata norton , how ministries can $^\ddagger$ |
| | 1004 | ∘ i [unused582] to gazing monterrey makeshift ca iata norton , how ministries can $^\ddagger$ |
| | 1005 | ∘ i [unused582] to gazing monterrey makeshift ca iata norton , how ministries can $^\ddagger$ |
| | 1006 | ∘ i [unused582] to gazing monterrey makeshift ca iata norton , how ministries can $^\ddagger$ |
| | 1007 | ∘ i [unused582] to gazing monterrey makeshift ca iata norton , how ministries can $^\ddagger$ |
| | 1008 | ∘ i [unused582] to gazing monterrey makeshift ca iata norton , how ministries can $^\ddagger$ |
| | 1009 | ∘ i [unused582] to gazing monterrey makeshift ca iata norton , how ministries can $^\ddagger$ |
| | 1250 | ∘ i want to prepare ##cchi my ca glove henan rouen how synthesized can [unused201] ##yya exam ? |
| | 1500 | ∘ i want to prepare for my ca cp exam , how transvaal can warmed ##yya exam ? |
| | 1750 | ∘ i want to prepare for my ca cp exam , how yun can get 2 exam ? |
| | 2000 | ∘ i want to prepare for my ca cp exam , how might can get 2 exam ? |

Source: i have only 2 months for my ca cpt exams how do i prepare?
Reference: i want to crack ca cpt in 2 months. how should i study?

**Multinomial Diffusion Does Not Decode Iteratively.** As presented in Tables 12 to 14, multinomial diffusion finishes the generation of most sentences in the first iteration and remains unchanged afterward, despite multiple iteration steps being allocated.

Table 13: A snapshot of all iterations for qualitative samples of test paraphrases generated from different diffusion models on QQP dataset. ‡ texts are truncated to fit into the table. Words are in lower case. <M> stands for mask states, and ## denotes the sub-word tokenization artifacts.

| | # Iter. | Decodes |
|---|---|---|
| **Source:** how can one increase concentration? 
 **Reference:** how can i improve my concentration? | | |
| Absorbing | 0 | ∘ <M> <M> <M> <M> <M> <M> <M> <M> |
| | 1 | ∘ <M> can i increase <M> <M> <M> <M> |
| | 2 | ∘ how can i increase concentration <M> <M> <M> |
| | 3 | ∘ how can i increase concentration in studying <M> |
| | 4 | ∘ how can i increase concentration in studying <M> |
| | 5 | ∘ how can i increase concentration in studying ? |
| RDM-absorbing | 0 | ∘ <M> <M> <M> <M> <M> <M> <M> <M> |
| | 1 | ∘ <M> <M> <M> <M> <M> concentration ? |
| | 2 | ∘ how <M> <M> <M> my concentration ? |
| | 3 | ∘ how <M> <M> increase my concentration ? |
| | 4 | ∘ how <M> i increase my concentration ? |
| | 5 | ∘ how can i increase my concentration ? |
| Multinomial | 0 | ∘ ##tly distances outline ##cera khmer curvature question ##tl |
| | 1 | ∘ how can i improve focus in concentration ? |
| | 2 | ∘ how can i improve focus in concentration ? |
| | 3 | ∘ how can i improve focus in concentration ? |
| | 4 | ∘ how can i improve focus in concentration ? |
| | 5 | ∘ how can i improve focus in concentration ? |
| RDM-multinomial | 0 | ∘ lungs ##down intensity cortes ##lden ufo oldies |
| | 1 | ∘ worker blurted i ##kal caledonia concentration ##vb |
| | 2 | ∘ how trait i ##kal my concentration ##vb |
| | 3 | ∘ how trait i increase my concentration ? |
| | 4 | ∘ how trait i increase my concentration ? |
| | 5 | ∘ how do i increase my concentration ? |
| DiffuSeq | 0 | ∘ skeptical coli ##zam gael erika calves wharf [unused791] ##pta vhf ##kley adoptive ‡ |
| | 10 | ∘ encompassing hesse informally campos cosmopolitan postmaster stabilization realised ‡ |
| | 100 | ∘ thump haitian i ##anov xiv ? ##] norris illuminated ##had kilometers disagreed [unused730]‡ |
| | 250 | ∘ fatal correlated trenton i ##anov exhibits ##] scandinavia 1934 plaza leveled 910 ‡ |
| | 500 | ∘ cessna i perez newark ? venezuelan regeneration 283 zhejiang ##hectares [PAD] ‡ |
| | 750 | ∘ johanna cessna i perez shriek ? [PAD] [PAD] [PAD] [PAD] rahman [PAD] [PAD] [PAD] [PAD] postmaster‡ |
| | 1000 | ∘ johanna 730 i improve terminals ? |
| | 1001 | ∘ johanna 730 i improve terminals ? |
| | 1002 | ∘ johanna 730 i improve terminals ? |
| | 1003 | ∘ johanna 730 i improve terminals ? |
| | 1004 | ∘ johanna 730 i improve terminals ? |
| | 1005 | ∘ johanna 730 i improve terminals ? |
| | 1006 | ∘ johanna 730 i improve terminals ? |
| | 1007 | ∘ johanna 730 i improve terminals ? |
| | 1008 | ∘ johanna 730 i improve terminals ? |
| | 1009 | ∘ johanna 730 i improve terminals ? |
| | 1250 | ∘ how do i improve concentration ? |
| | 1500 | ∘ how do i improve concentration ? |
| | 1750 | ∘ how do i improve concentration ? |
| | 2000 | ∘ how do i improve concentration ? |

This unexpected behavior is due to the formulation of its original backward process, which is of the form as Equation 14 (copied here for convenience),

$$
p_{\boldsymbol{\theta}}(\mathbf{x}_{t-1}|\mathbf{x}_t)
$$
$$
= \frac{\alpha_t \mathbf{x}_t \odot f(\mathbf{x}_t; \boldsymbol{\theta}) + \frac{1}{K}\beta_t(1-\alpha_{t-1})\mathbf{x}_t + \frac{1}{K}(1-\beta_t)\alpha_{t-1}f(\mathbf{x}_t; \boldsymbol{\theta}) + \frac{1}{K^2}(1-\beta_t)(1-\alpha_{t-1})\mathbf{1}}{\alpha_t \mathbf{x}_t^\top f(\mathbf{x}_t; \boldsymbol{\theta}) + \frac{1}{K}(1-\alpha_t)}.
$$

Note that $f(\cdot; \boldsymbol{\theta})$ is a softmax probability vector output from a Transformer model. At the initial iteration, $\alpha_t$ is very close to zero, and the Transformer prediction $f(\cdot; \boldsymbol{\theta})$ has the chance to come into play and denoise to a certain degree. But when the process moves onward, $\alpha_t$ becomes larger, which will soon make the first term dominate significantly over the others since all the other terms are scaled down by $1/K$. Since the vocabulary size $K$ is usually large in text generation tasks (usually larger than 10K), this would make all the other terms very close to zero. In this case, the backward

Table 14: A snapshot of all iterations for generated translates from different diffusion models on `IWSLT14 DE-EN` benchmark. Words are in lower case. <M> stands for mask states, and ## denotes the sub-word tokenization artifacts.

| | # Iter. | Decodes |
|---|---|---|
| **Source:** alleine dieser flughafen hat eine fläche von 100 quadratkilometern . | | |
| **Reference:** this airport alone covers more than 100 square kilometers . | | |
| Absorbing | 0 | ○ <M> <M> <M> <M> <M> <M> <M> <M> <M> <M> <M> <M> <M> <M> |
| | 1 | ○ <M> <M> <M> <M> <M> has an an <M> of <M> <M> miles <M> |
| | 2 | ○ <M> <M> <M> <M> <M> has an an <M> of <M> <M> miles <M> |
| | 3 | ○ <M> <M> air## <M> <M> has an an <M> of <M> <M> miles <M> |
| | 4 | ○ <M> <M> air## <M> <M> has an an <M> of <M> square miles <M> |
| | 5 | ○ <M> this air## port alone has an an <M> of <M> square miles <M> |
| | 6 | ○ <M> this air## port alone has an an <M> of <M> square miles <M> |
| | 7 | ○ <M> this air## port alone has an an <M> of <M> square miles . |
| | 8 | ○ <M> this air## port alone has an an <M> of 100 square miles . |
| | 9 | ○ <M> this air## port alone has an an <M> of 100 square miles . |
| | 10 | ○ and this air## port alone has an an area of 100 square miles . |
| RDM-absorbing | 0 | ○ <M> <M> <M> <M> <M> <M> <M> <M> <M> <M> <M> <M> <M> |
| | 1 | ○ <M> <M> air## <M> <M> <M> <M> <M> <M> <M> square <M> <M> |
| | 2 | ○ <M> <M> air## <M> <M> <M> <M> <M> <M> <M> square <M> . |
| | 3 | ○ <M> <M> air## <M> alone <M> <M> area <M> 100 square kilometers . |
| | 4 | ○ <M> <M> air## port <M> <M> an <M> of <M> square kilometers . |
| | 5 | ○ <M> <M> air## <M> <M> <M> <M> area of 100 square kilometers . |
| | 6 | ○ <M> this air## port <M> <M> an area of <M> square kilometers . |
| | 7 | ○ <M> this air## port alone <M> an area of 100 square kilometers . |
| | 8 | ○ <M> this air## port <M> has an area of 100 square kilometers . |
| | 9 | ○ <M> this air## port alone has an area of 100 square kilometers . |
| | 10 | ○ so this air## port alone has an area of 100 square kilometers . |
| Multinomial | 0 | ○ eher spending des## vagina drin production mili## inven## primi## open## freiheit sit schlüssel search |
| | 1 | ○ alone alone air## air## port has a a area of 100 square miles . |
| | 2 | ○ alone alone air## air## port has a a area of 100 square miles . |
| | 3 | ○ alone alone air## air## port has a a area of 100 square miles . |
| | 4 | ○ alone alone air## air## port has a a area of 100 square miles . |
| | 5 | ○ alone alone air## air## port has a a area of 100 square miles . |
| | 6 | ○ alone alone air## air## port has a a area of 100 square miles . |
| | 7 | ○ alone alone air## air## port has a a area of 100 square miles . |
| | 8 | ○ alone alone air## air## port has a a area of 100 square miles . |
| | 9 | ○ alone alone air## air## port has a a area of 100 square miles . |
| | 10 | ○ alone alone air## air## port has a a area of 100 square miles . |
| RDM-multinomial | 0 | ○ beschreiben denk## architect mittleren words alism grou## hilft atoms pus he## jähri## enti## ball## generally |
| | 1 | ○ expe## standing nahme cted baum katastrop## bares tion later colle## haufen 100 anstatt zy . |
| | 2 | ○ cognitive standing natürlich cted ution ity bares an later aus## informa## 100 square zy . |
| | 3 | ○ cognitive standing llig port wieder oth has an later erhalten saal 100 square kilometers . |
| | 4 | ○ crime standing air## port spending imag## has an area incredible of 100 square prototyp## . |
| | 5 | ○ crime standing air## port alone psychi## edi## an area oder of 100 square prototyp## . |
| | 6 | ○ starke standing air## port alone psychi## armut an area out of 100 square kilometers . |
| | 7 | ○ starke that air## port alone psychi## armut an area out of 100 square kilometers . |
| | 8 | ○ starke that air## port alone has armut an area out of 100 square kilometers . |
| | 9 | ○ and that air## port alone has got an area out of 100 square kilometers . |
| | 10 | ○ and that air## port alone has got an area out of 100 square kilometers . |

transition distribution degenerates to

$$p_{\boldsymbol{\theta}}(\mathbf{x}_{t-1}|\mathbf{x}_t) \approx \frac{\alpha_t \mathbf{x}_t \odot f(\mathbf{x}_t; \boldsymbol{\theta})}{\alpha_t \mathbf{x}_t^\top f(\mathbf{x}_t; \boldsymbol{\theta})} = \mathbf{x}_t.$$

That is, what multinomial diffusion does after the initial steps is merely copying previous states, and hence the sequence mostly remains unchanged. The only chance for the multinomial diffusion processes to decode is at the initial stage; after that, the model would get stuck in the current state and cannot escape. This explains why multinomial diffusion does not behave like typical iterative processes.

Our *reparameterization* does not suffer from these issues. Thanks to Equation 2, the developed reparameterized formulation alleviates the need to normalize all terms together; instead, it divides different terms into two cases, which are then normalized separately. This avoids the possibility that different terms are affected by their relative scales. The resulting behavior is much more expected and leads to better generation quality.

**The Slow Convergence of Continuous Diffusion.** In contrast to discrete diffusion models that can perform generation in 10 steps or fewer, continuous diffusion usually requires thousands of steps to decode a decent sample. We hypothesize that this is due to two reasons: (1) the noisy and slow Gaussian diffusion over token embeddings by design; (2) furthermore, many diffusing steps are required to emit a significant change over token states due to the rounding operation. We provide empirical evidence for our hypothesis by zooming in to inspect the generation process. As can be seen in Tables 12 and 13, many consecutive steps in continuous diffusion (1000~1009 iteration) do not modify the decode at all, even when it is not converged yet, leading to a potential waste of computation.

**Vanilla Absorbing Diffusion Cannot Fix Previous Errors.** While vanilla absorbing diffusion performs decoding more steadily, it also suffers from some issues. As shown in Tables 12 to 14, since all tokens are predicted independently conditional on the source input, there is some chance for the model to decode multiple identical tokens simultaneously. However, in vanilla absorbing diffusion, such decoding errors cannot be fixed. To see this, note that its backward transition formulation can be written as Equation 10, which is copied here for convenience,

$$p_{\boldsymbol{\theta}}(\mathbf{x}_{t-1}|\mathbf{x}_t) = \begin{cases} \left(1 - \frac{1-\alpha_{t-1}}{1-\alpha_t}\right) f(\mathbf{x}_t; \boldsymbol{\theta}) + \frac{1-\alpha_{t-1}}{1-\alpha_t} q_{\text{noise}}, & \text{if } \mathbf{x}_t = [M] \\ \mathbf{x}_t, & \text{if } \mathbf{x}_t \neq [M]. \end{cases}$$

Under this backward formulation, once a token is decoded, it will stay in the state thereafter and does not have the chance to be re-predicted again. As a result, vanilla absorbing diffusion processes cannot fix previously made errors.

This issue can be alleviated by RDMs, which employ a more generic formulation as follows,

$$p_{\boldsymbol{\theta}}(\mathbf{x}_{t-1}|\mathbf{v}_{t-1}, \mathbf{x}_t) = \begin{cases} v_{t-1}^{(1)} \mathbf{x}_t + \left(1 - v_{t-1}^{(1)}\right) q_{\text{noise}}, & \text{if } b_t = 1 \\ v_{t-1}^{(2)} f(\mathbf{x}_t; \boldsymbol{\theta}) + \left(1 - v_{t-1}^{(2)}\right) q_{\text{noise}}(\mathbf{x}_t), & \text{if } b_t = 0. \end{cases}$$

Here $b_t$ is a binary variable indicating whether $\mathbf{x}_t$ is denoised or not, and $v_{t-1}^{(1)}$ can be either 1 or 0, depending on the strategy (§4.3). Therefore, in RDMs, we can allow decoded tokens to be rolled back to noisy states by setting $v_{t-1}^{(1)} = 0$ (e.g., these repetitive tokens might receive lower model scores than the others, which can be recognized as low-confidence outputs in Equation 7). An example can be found in Table 12, where the decoded repetitive tokens `months months` at 3-th iteration are then re-masked at the next iteration.

