# OpenReview forum: "A Reparameterized Discrete Diffusion Model for Text Generation"
_ICLR.cc/2024/Conference — Submitted to ICLR 2024_

### Official Review · Reviewer_Z8Rn · 2023-10-27

**Soundness:** 4 excellent
**Presentation:** 4 excellent
**Contribution:** 4 excellent
**Rating:** 8
**Confidence:** 4

**Summary:**

This paper develops a novel framework for discrete diffusion models. In particular, they develop an adaptive routing strategy that routes tokens to the denoised state only if the router outputs high scores instead of uniformly processing all the tokens.  And extensive experiments have been conducted to evaluate the text generation capability of their model, demonstrating significant improvements over existing diffusion models

In my opinion, this is a novel work that divides the generative process into two processes, consisting of a noise token and an unnoise token. Generally, if the state is a noise token, the generative process selects to denoise with a neural network. If the state is an unnoise token, the generative process selects to add noise. Compared with the base diffusion model, this method reduces the generative difficulty of the generative network, which only needs to take denoise tasks.   From another principle, the method can be seen as a decoupling method that makes the nework not need to learn two tasks (add noise and denoise) at the same time.

**Strengths:**

1) This paper proposes a novel framework for the discrete diffusion model, and it is well written and easy to understand, although it has many mathematical

2) This paper provides us with new insight into the diffusion model.

3) This work gets effective performance with extensive experiments.

**Weaknesses:**

Can you take more experiments on different length language datasets that is aimed at exploring  model performance boundaries? In my opinion, the diffusion model hard to generates a longer sequence than the auto-agressive language model

**Questions:**

Is it simple to adapt this method to other diffusion models, such as the Gaussian diffusion model? If not, why?


why the recursive computation for b_t is effective; I'm confused on it.

---

> ### Author Response · Authors · 2023-11-22
>
> Thanks for your feedback and thoughtful comments! Here are our responses:
>
> > **Q1:** Can you take more experiments on different length language datasets that is aimed at exploring model performance boundaries? In my opinion, the diffusion model hard to generates a longer sequence than the auto-agressive language model
> >
>
> **A1:** We conduct additional experiments to explore the capabilities of RDM in open-ended text generation, particularly on Wikitext-103, which is an established dataset for language models. Please refer to the **general response #3** as well as Appendix G.3 in our updated manuscript for details.
>
> > **Q2:** Is it simple to adapt this method to other diffusion models, such as the Gaussian diffusion model? If not, why?
> >
>
> **A2:** Thanks for your input. Our reparameterization, specifically derived for discrete diffusion models, presents challenges in direct application to Gaussian diffusion models due to their inherently different formulations (Gaussian diffusion models operate on continuous values). However, the underlying concept of separating the transition process into routing and denoising mechanisms could be beneficial in broader diffusion model classes. Exploring this reparameterization in a wider range of diffusion processes is an interesting direction for future research.
>
> > **Q3:** why the recursive computation for b_t is effective; I'm confused on it.
> >
>
> **A3:** $b_t$ is used to check whether the current token state $x_t$ is noiseless (i.e., equal to the ground-truth input $x_0$), which depends on $x_0$, but $x_0$ is not available during backward generation. To calculate $b_t$ without the presence of $x_0$, we propose this technique by initializing a bit vector $b_t$ at the start of backward generation. As tokens within the sequence are iteratively denoised during backward generation, this bit vector records denoised token positions and is sequentially updated to represent the noise status of each token. Therefore, the determination of $b_t$ can be accomplished independently of the direct presence of $x_0$.

---

### Official Review · Reviewer_shFm · 2023-10-30

**Soundness:** 3 good
**Presentation:** 3 good
**Contribution:** 2 fair
**Rating:** 5
**Confidence:** 4

**Summary:**

This paper presents a reparameterization approach for discrete diffusion models applied to text generation tasks. By introducing two additional variables to process noisy and original inputs separately, the proposed method aims to improve the performance of discrete diffusion models. The experiments are conducted on machine translation, question generation, and paraphrasing tasks, demonstrating improvements over baseline discrete diffusion models.

**Strengths:**

- The paper attempts to provide a new perspective on discrete diffusion models by introducing a reparameterization approach, which could potentially lead to better results in text generation tasks.

- The experiments conducted cover a variety of text generation tasks and show improvements over the baseline discrete diffusion models.

**Weaknesses:**

- The motivation behind the proposed method is not clearly explained, and the paper lacks a solid theoretical foundation to support the reparameterization approach. The reasoning behind dividing the diffusion of tokens into two conditions by comparing with the original input is not well-justified. Additionally, the paper does not provide a thorough theoretical analysis to support the claim that the proposed method can lead to better results compared to the original diffusion process.

- The proposed method appears to borrow heavily from masked language modeling techniques that have already been utilized in non-autoregressive text generation works, such as CMLM, DisCo, and others. Furthermore, existing discrete diffusion models like improved VQ-Diffusion have proposed advanced models that consider each token separately, making the contribution of the proposed method less novel and impactful.

- The performance improvements demonstrated by the proposed method over the discrete diffusion baseline are not consistent across different tasks and metrics. This raises concerns about the generalizability and robustness of the proposed method for various text generation tasks.

- The choice of baselines and metrics for comparison in the experiments could be more appropriate. For example, in Figure 2 (b), the comparison with Diffuseq, a continuous baseline, is not suitable, and the performance of Diffuseq is not well-explained. Moreover, the paper does not follow the experimental settings of CMLM, which achieves higher performance in their original paper.

- The paper does not provide an in-depth analysis of the discrepancy between training and inference introduced by the proposed method. The decoding method presented is similar to existing mask-predict and easy-first policies applied in non-autoregressive or diffusion models, raising questions about the novelty and effectiveness of the proposed approach.

- The paper does not include a comprehensive ablation study to demonstrate the effectiveness of the proposed method and its individual components. A detailed ablation study could provide insights into the contribution of each component to the overall performance improvements.

- The experiments in the paper focus mainly on conditional text generation tasks, where a source input is provided. It remains unclear whether the proposed reparameterization approach can also enhance performance in other text generation tasks, such as unconditional language modeling, limiting the potential impact and generalizability of the method.

In light of the above weaknesses, I recommend rejecting this paper as it has limited contribution to the field, lacks a clear motivation, and its improvements over the baselines are not consistent or substantial.

**Questions:**

Please refer to weaknesses for the questions.

---

> ### Author Response · Authors · 2023-11-22
>
> Thanks for your feedback and thoughtful comments! Here are our responses:
>
> > **Q1:** The motivation behind the proposed method is not clearly explained, and the paper lacks a solid theoretical foundation to support the reparameterization approach. The reasoning behind dividing the diffusion of tokens into two conditions by comparing with the original input is not well-justified. Additionally, the paper does not provide a thorough theoretical analysis to support the claim that the proposed method can lead to better results compared to the original diffusion process.
> >
>
> **A1:** The developed reparameterization technique provides a strictly equivalent formulation to the vanilla discrete diffusion process. The mechanism that divides the diffusion of tokens into two conditions stems directly from this reparameterization. This theoretical grounding is elaborated upon in our paper. We further demonstrate in our experiments (especially Section 5.3 and Appendix G) that the disentanglement between the routing and denoising mechanism facilitates the separate parameterization of these two modules and makes it easier for the neural network to learn to denoise text and mitigate the risk of degeneration.
>
> > **Q2:** The proposed method appears to borrow heavily from masked language modeling techniques that have already been utilized in non-autoregressive text generation works, such as CMLM, DisCo, and others. Furthermore, existing discrete diffusion models like improved VQ-Diffusion have proposed advanced models that consider each token separately, making the contribution of the proposed method less novel and impactful.
> >
>
> **A2:** Masked language models can be considered as a particular instance of general discrete diffusion models, namely absorbing diffusion. Our main technical contribution involves re-formulating existing discrete diffusion processes, which decouples between routing and denoising mechanisms during backward generation. To the best of our knowledge, these results are new in the area of text generation models and have demonstrated strong performance across various tasks. Regarding ``considering each token separately'', it is a common assumption in most discrete diffusion models (if the distribution among different tokens is not factorized, it would become intractable since the size of its support would be exponential in sequence length). This assumption is commonly made to ensure the feasibility of the model.
>
> > **Q3:** The performance improvements demonstrated by the proposed method over the discrete diffusion baseline are not consistent across different tasks and metrics. This raises concerns about the generalizability and robustness of the proposed method for various text generation tasks.
> >
>
> **A3:** The generation quality among discrete diffusion baselines has been extensively compared in Tables 1 and 3, which shows that the reparameterized variants all outperform vanilla discrete diffusion baselines consistently by a large margin.
>
> > **Q4:** The choice of baselines and metrics for comparison in the experiments could be more appropriate. For example, in Figure 2 (b), the comparison with Diffuseq, a continuous baseline, is not suitable, and the performance of Diffuseq is not well-explained. Moreover, the paper does not follow the experimental settings of CMLM, which achieves higher performance in their original paper.
> >
>
> **A4:** Including continuous diffusion models like DiffuSeq in Figure 2(b) validates our method's effectiveness and efficiency against both discrete and continuous models. Regarding the performance of DiffuSeq, it often produces noisy results early on and thus leads to non-sense text with near zero BLEU score when #steps < 250, requiring substantially longer runtime to obtain decent results. In terms of the experimental settings of CMLM, our goal was to develop a model based on non-autoregressive diffusion probabilistic processes and remove any kinds of dependencies on AR approaches (unlike CMLM, whose results come from the variants distilled from autoregressive Transformer large). Hence, we opted to train the diffusion model on the original WMT14 EN-DE dataset, rather than the distilled split.

---

> > ### Author Response · Authors · 2023-11-22
> >
> > > **Q5:** The paper does not provide an in-depth analysis of the discrepancy between training and inference introduced by the proposed method. The decoding method presented is similar to existing mask-predict and easy-first policies applied in non-autoregressive or diffusion models, raising questions about the novelty and effectiveness of the proposed approach.
> > >
> >
> > **A5:** Our developed reparameterization, which disentangles the underlying token selection (routing) from the token prediction (denoising) process during backward sampling, allows for integration with the mask-predict heuristic across various discrete diffusion models. This includes both multinomial and absorbing diffusion, not limited to masking-based models. As demonstrated in Table 1 and Figures 2a, 3a, and 3b, this integration results in a significantly enhanced generation quality, reaching levels comparable to auto-regressive baselines. We hope that this clarification underscores the unique contributions of our proposed methodology relative to the existing body of work.
> >
> > > **Q6:** The paper does not include a comprehensive ablation study to demonstrate the effectiveness of the proposed method and its individual components. A detailed ablation study could provide insights into the contribution of each component to the overall performance improvements.
> > >
> >
> > **A6:** We have conducted detailed ablation studies in Section 5.3 (including Table 2, Figure 2a, 3a, 3b) to investigate the effect of each component.
> >
> > > **Q7:** The experiments in the paper focus mainly on conditional text generation tasks, where a source input is provided. It remains unclear whether the proposed reparameterization approach can also enhance performance in other text generation tasks, such as unconditional language modeling, limiting the potential impact and generalizability of the method.
> > >
> >
> > **A7:** We conduct additional experiments to explore the capabilities of RDM in open-ended text generation, particularly on Wikitext-103, which is an established dataset for language models. Please refer to the **general response #3** as well as Appendix G.3 in our updated manuscript for details.

---

> > > ### Comment · Reviewer_shFm · 2023-11-23
> > >
> > > I have carefully read the authors' responses as well as other reviews. I appreciate the clarification regarding the theoretical foundation and motivation behind the proposed reparameterization approach. However, the responses do not fully alleviate the concerns raised in the original review.
> > >
> > > - The claim that the reparameterization is strictly equivalent to the vanilla discrete diffusion process needs further elaboration and a more thorough theoretical analysis. While the authors assert that disentangling routing and denoising facilitates learning, the response lacks concrete evidence or a compelling argument to support this assertion.
> > >
> > > - The explanation regarding the relationship between masked language models and discrete diffusion models is noted, but the novelty and impact of the proposed method relative to existing works remain questionable. The response acknowledges the common assumption of considering each token separately, but it does not sufficiently address the perception that the contribution is less novel and impactful compared to advanced models like improved VQ-Diffusion.
> > >
> > > - The response on performance consistency across tasks is not entirely convincing. The concerns raised about the variability in performance improvements across different tasks and metrics still remain.
> > >
> > > In light of the above, I maintain my original recommendation for weak rejection.

---

### Official Review · Reviewer_7X9e · 2023-11-01

**Soundness:** 3 good
**Presentation:** 2 fair
**Contribution:** 2 fair
**Rating:** 3
**Confidence:** 4

**Summary:**

This research delves into discrete diffusion probabilistic models, specifically focusing on their applications in conditional natural language generation. The study introduces an innovative and equivalent approach for sampling anf training from discrete diffusion processes. This novel framework provides a fresh perspective on the generation process within these models and incorporates more effective training and decoding techniques. Extensive experiments are conducted to assess the text generation capabilities of this model.

**Strengths:**

This paper introduces a novel framework for discrete diffusion probabilistic models. It presents a comprehensive array of experiments conducted on conditional NLP datasets, showcasing significant enhancements compared to existing diffusion models.

**Weaknesses:**

The main critique of this paper centers on its narrow experimental scope, concentrating solely on conditional text generation tasks such as machine translation, question generation, and paraphrasing. This focus is somewhat limiting, particularly given the absence of experiments on unconditional text data. This omission is notable, especially considering the extensive body of prior work in this area, including significant contributions by Austin et al. (2021) and Hoogeboom et al. (2022b), who have extensively explored these scenarios.

This paper, although introducing a new formulation and reparameterization of discrete diffusion models, primarily builds upon existing frameworks. The use of discrete diffusion processes for text generation is not entirely novel, and the proposed modifications could be considered incremental improvements to established methods.

**Questions:**

1. How does your proposed method perform on established conditional NLP datasets, such as enwik8 and text8, as previously tested in well-regarded papers (Hoogeboom et al., 2021 and Austin et al., 2021)?

2. To what degree does the quality of your text generation process rely on the choice of your training objective during the training phase? If you were to employ the loss functions utilized in Hoogeboom et al., 2021 and Austin et al., 2021 to train your diffusion model, how would this impact the quality of the generated text?

---

> ### Author Response · Authors · 2023-11-22
>
> Thanks for your feedback and thoughtful comments! Here are our responses:
>
> > **Q1:** How does your proposed method perform on established conditional NLP datasets, such as enwik8 and text8, as previously tested in well-regarded papers (Hoogeboom et al., 2021 and Austin et al., 2021)?
> >
>
> **A1:** We conduct additional experiments to explore the capabilities of RDM in open-ended text generation, particularly on Wikitext-103, which is an established dataset for language models. Please refer to the **general response #3** as well as Appendix G.3 in our updated manuscript for details.
>
> > **Q2:** To what degree does the quality of your text generation process rely on the choice of your training objective during the training phase? If you were to employ the loss functions utilized in Hoogeboom et al., 2021 and Austin et al., 2021 to train your diffusion model, how would this impact the quality of the generated text?
> >
>
> **A2:** We have conducted the ablation study, detailed in Figures 2a, 3a, and 3b, to systematically evaluate the effect of separate training and decoding improvements. Our results demonstrate that employing reparameterized decoding alone (i.e., applying the loss functions utilized in Hoogeboom et al., 2021 and Austin et al., 2021 to train the diffusion model, captioned as Improved Decoding), already significantly improves performance compared to vanilla baselines.

---

### Official Review · Reviewer_uT11 · 2023-11-01

**Soundness:** 4 excellent
**Presentation:** 4 excellent
**Contribution:** 3 good
**Rating:** 6
**Confidence:** 5

**Summary:**

The paper addresses the challenge of applying discrete diffusion probabilistic models to natural language generation. The authors propose a novel reparameterized discrete diffusion model (RDM) by re-examining the sampling process from discrete diffusion models. They introduce a route-and-denoise process that includes a stochastic routing mechanism, which makes the training more efficient by simplifying the training objective to a reweighted standard cross-entropy loss. This new family of models, RDMs, demonstrates significant improvements in terms of effectiveness and flexibility over existing diffusion models in text generation tasks.

The RDMs achieve high-quality text generation by offering a more effective training and decoding process, which can be highly flexible and adaptive. The model's performance is evaluated across various text generation benchmarks, showing superior results over both existing discrete and continuous diffusion models while operating several orders of magnitude faster. This work pushes the boundaries of non-autoregressive text generation, providing a fresh perspective on the discrete diffusion approach and opening pathways for further research and application in more complex language tasks.

**Strengths:**

Advantages:
 - Improved Training and Decoding Techniques Allows Efficient and Effective Non-autoregressive Generation: The RDM incorporates improved training and decoding strategies that significantly enhance performance over vanilla baselines. The derived loss objective, formulated as a reweighting cross-entropy function, and the discriminative routing mechanism for decoding are highlighted as key contributors to this performance boost.
 - Solid Performance on Machine Translation Experiments: RDM achieves a milestone-level performance as a diffusion-based Machine Translation model with either competititive or superior performance compared to previous Non-autoregressive translation models.

**Weaknesses:**

Disadvantages:
 - Limited Comparison: I always think the comparison between text diffusion models and other important non-autoregressive models is important. This is not only because of the concerns for a fair and thorough comparison. This is more because of the fact that such non-autoregressive models **are** text diffusion models that has a diffusion process defined as operating the discrete tokens. Levenshtein Transformer, for example, is cited yet not quite compared in many of the experiments, which limits the soundness of the experiments. I encourage the authors to add them in (many of the results can actually be directly borrowed from their original paper I think).

- Limited Choice of Tasks that Raises Concerns of Overclaiming: This method is now mostly tested on semantically deterministic tasks like Machine Translation. There's still a very huge gap between getting the model to function on MT tasks and getting it to produce diverse, informative outputs on open-domain text generation tasks. With this concern, I would tend to suggest the authors to claim smaller, to restrict the claim to be a diffusion-based machine translation model instead of claiming it as a diffusion-based text generation model.

**Questions:**

Is it possible to also conduct a running time study to compare RDM with other non-autoregressive models? Currently only previous diffusion sequence models and CausalLM baselines are compared. This comparison can sometimes be tricky because many irrelevant factors like the quality of implementation also impacts the running time. But I would be more convinced about the practical value if I see such results in the experiment section (since the model itself is already falling short in terms of #refinements compared to LevT, but if each refinement with RDM is computationally cheaper, there's still a chance for RDM to be faster)

---

> ### Author Response · Authors · 2023-11-22
>
> Thanks for your feedback and thoughtful comments! Here are our responses:
>
> > **Q1:**  I always think the comparison between text diffusion models and other important non-autoregressive models is important. This is not only because of the concerns for a fair and thorough comparison. This is more because of the fact that such non-autoregressive models **are** text diffusion models that has a diffusion process defined as operating the discrete tokens. Levenshtein Transformer, for example, is cited yet not quite compared in many of the experiments, which limits the soundness of the experiments. I encourage the authors to add them in (many of the results can actually be directly borrowed from their original paper I think).
> >
>
> **A1:** We appreciate your insightful suggestion regarding the comparison between diffusion models and other non-autoregressive models. We agree that many such non-autoregressive models can be cast as a diffusion process, where transformations are applied to discrete tokens, and the model learns to reverse the operation, inducing a generative model. In our study, we have primarily compared RDMs against the established non-autoregressive model CMLM. This comparison highlights the superior performance of RDMs over previous non-autoregressive baselines. The Levenshtein Transformer employs a rather distinct architecture with multiple output heads dedicated to learning deletion and insertion operations, making direct comparisons challenging. However, we have included the comparison in terms of experimental results in the updated Table 1, where RDMs outperform the Levenshtein Transformer by a large margin (e.g., 27.59 vs. 25.20 BLEU on the WMT14 EN-DE benchmark).
>
> > **Q2:** This method is now mostly tested on semantically deterministic tasks like Machine Translation. There's still a very huge gap between getting the model to function on MT tasks and getting it to produce diverse, informative outputs on open-domain text generation tasks. With this concern, I would tend to suggest the authors to claim smaller, to restrict the claim to be a diffusion-based machine translation model instead of claiming it as a diffusion-based text generation model.
> >
>
> **A2:** We conduct additional experiments to explore the capabilities of RDM in open-ended text generation, particularly on Wikitext-103, which is an established dataset for language models. Please refer to the **general response #3** as well as Appendix G.3 in our updated manuscript for details.
>
> > **Q3:** Is it possible to also conduct a running time study to compare RDM with other non-autoregressive models? Currently only previous diffusion sequence models and CausalLM baselines are compared. This comparison can sometimes be tricky because many irrelevant factors like the quality of implementation also impacts the running time. But I would be more convinced about the practical value if I see such results in the experiment section (since the model itself is already falling short in terms of #refinements compared to LevT, but if each refinement with RDM is computationally cheaper, there's still a chance for RDM to be faster)
> >
>
> **A3:** All models are implemented with the same codebase (FairSeq). We observe the primary factor affecting decoding runtime among non-autoregressive baselines and our diffusion models is the number of Transformer decoder calls. Consequently, the number of iteration steps (equivalent to the number of Transformer decoder calls) faithfully reflects the actual decoding speed.
>
> For instance, the comparisons against non-autoregressive baselines are provided here (and also in Appendix G.2 of our revised manuscript): CMLM and RDM-absorbing take 44.2-45.0s and 44.0-45.3s, respectively, to decode the WMT14 EN-DE test set on one NVIDIA GeForce RTX 3090 GPU with 50 batch size and 16 iteration steps, averaged by 10 runs. Multinomial/reparameterized-multinomial diffusion models take ~46.8s; in general, the overhead, which mainly involves token-wise masking/assignment operations, is marginal compared to that of Transformer decoder calls.

---

### Author Response · Authors · 2023-11-22
**Summary of Updates**

We thank the reviewers for their insightful suggestions and comments. We revised the paper accordingly, and here is a summary of updates:

1. In **Section 5.1**, we update Table 1 to include more non-autoregressive baselines in comparison.
2. In **Appendix G.2**, we provide more details on the decoding runtime comparison between non-autoregressive models (in particular, CMLM) and our method.
3. In **Appendix G.3**, we conduct additional experiments on open-ended text generation to further explore the capabilities of RDMs, particularly on the Wikitext-103 dataset. Due to constraints in computational resources, a comprehensive evaluation is challenging at this point; however, preliminary results (Appendix G.3) suggest that RDMs greatly improve vanilla discrete diffusion processes and perform competitively against auto-regressive language models. While still lagging behind auto-regressive language models in terms of coherence and MAUVE scores, RDMs demonstrate a notable ability to generate coherent and more varied open-ended text. These results highlight the generality and effectiveness of our method in improving diffusion models for text generation.

---

### Meta-Review · Area_Chair_4b5Z · 2023-12-21

**Metareview:**

This paper proposes an equivalent formulation of the sampling from discrete diffusion processes  and leverages this insight to develop a family of reparameterized discrete diffusion models. The proposed framework features more effective training and decoding techniques for different discrete diffusion models. The proposed framework enhances training and decoding techniques across various discrete diffusion models. Experiments on machine translation demonstrate performance improvement over existing diffusion models. However, the reviewers have the following major concerns: (1) Inconsistent performance improvements across tasks and metrics raise doubts about the proposed method's generalizability; (2) lack of comparison with other non-autoregressive models; (3) lack of evaluation on more diverse benchmarks beyond machine translation, e.g., unconditional language modeling; (4) lack of sufficient ablation study. Despite author rebuttal and reviewer discussion, the paper does not get sufficient support. Thus I recommend rejection.

**Justification For Why Not Higher Score:**

(1) Inconsistent performance improvements across tasks and metrics raise doubts about the proposed method's generalizability; (2) lack of comparison with other non-autoregressive models; (3) lack of evaluation on more diverse benchmarks beyond machine translation, e.g., unconditional language modeling; (4) lack of sufficient ablation study.

**Justification For Why Not Lower Score:**

N/A

---

### Decision · Program_Chairs · 2024-01-16

Reject